# Dual Prototype Evolving for Test-Time Generalization of Vision-Language Models

**Ce Zhang    Simon Stepputtis    Katia Sycara    Yaqi Xie**
School of Computer Science, Carnegie Mellon University
{cezhang, sstepput, katia, yaqix}@cs.cmu.edu

## Abstract

Test-time adaptation, which enables models to generalize to diverse data with unlabeled test samples, holds significant value in real-world scenarios. Recently, researchers have applied this setting to advanced pre-trained vision-language models (VLMs), developing approaches such as test-time prompt tuning to further extend their practical applicability. However, these methods typically focus solely on adapting VLMs from a single modality and fail to accumulate task-specific knowledge as more samples are processed. To address this, we introduce Dual Prototype Evolving (DPE), a novel test-time adaptation approach for VLMs that effectively *accumulates* task-specific knowledge from *multi-modalities*. Specifically, we create and evolve two sets of prototypes—textual and visual—to progressively capture more accurate multi-modal representations for target classes during test time. Moreover, to promote consistent multi-modal representations, we introduce and optimize learnable residuals for each test sample to align the prototypes from both modalities. Extensive experimental results on 15 benchmark datasets demonstrate that our proposed DPE consistently outperforms previous state-of-the-art methods while also exhibiting competitive computational efficiency. Code is available at https://github.com/zhangce01/DPE-CLIP.

## 1 Introduction

Although deep learning models have achieved great success in various machine learning tasks [48, 49], they often suffer from significant performance degradation due to distribution shifts between the training data from the source domain and the testing data from the target domain [34, 64, 15]. To address this challenge, a number of works [22, 58, 65] adopt the transductive learning principle, assuming access to both labeled source data and unlabeled target data—a scenario known as the *domain adaptation* setting. However, this setting contrasts with most practical scenarios, where we only have access to a well-trained model and cannot re-access the source data due to privacy or data retention policies. In response, researchers have proposed *test-time adaptation*, which leverages only the unlabeled target data stream to adapt the model to out-of-distribution domains [79, 60, 62].

Recently, large-scale vision-language models (VLMs), such as CLIP [46] and ALIGN [25], have garnered increasing attention in the research community. These models, pre-trained on massive web-scale datasets, exhibit remarkable zero-shot capabilities and open-world visual understanding [46, 70, 74, 32]. While the large-scale pre-trained (source) datasets like LAION-5B [50] are accessible, it is impractical for individuals to train on them due to their immense size. Consequently, adapting VLMs to downstream tasks via efficient fine-tuning with limited annotated samples from the target domain has become a focus of recent research [85, 84, 81, 71]. However, although these methods have proven effective, they pose a significant limitation: they assume the availability of annotated samples from the target domain, which is often not practical in real-world scenarios. This constraint hinders the broader deployment of VLMs in diverse and dynamic environments [20, 21, 47, 75].

38th Conference on Neural Information Processing Systems (NeurIPS 2024).

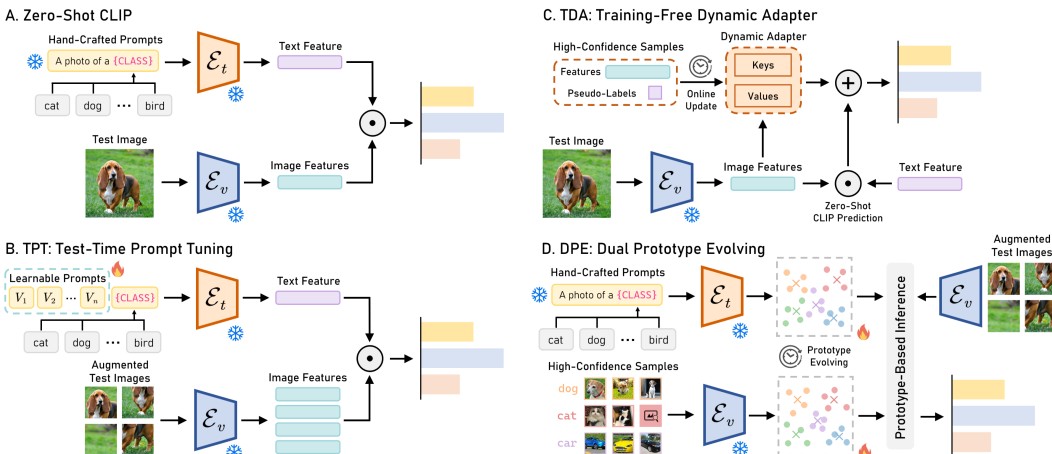

Figure 1: **Comparison of our DPE with zero-shot CLIP [46], TPT [54], and TDA [27]**. We denote CLIP's parallel textual and visual encoders as $\mathcal{E}_t$ and $\mathcal{E}_v$, respectively. While previous methods solely adapt the CLIP model from a single modality, we design our DPE to evolve prototypes from both textual and visual modalities to progressively capture more accurate multi-modal representations for target classes during test time.

To address the label scarcity problem in practice, a number of approaches apply the *test-time adaptation* setting to the domain of adapting VLMs to downstream tasks, as shown in Figure 1. Specifically, Shu *et al*. [54] propose test-time prompt tuning to learn an adaptive prompt for each individual sample in the test data stream to enhance CLIP's zero-shot generalizability to out-of-distribution domains. Building on TPT, DiffTPT [13] incorporates diffusion-based data augmentations to facilitate more effective prompt tuning during test time. More recently, Karmanov *et al*. [27] propose an alternative training-free dynamic adapter approach to establish dynamic visual caches with the unlabeled test samples.

However, we recognize that existing works overlook the following inherent properties of *test-time adaptation* in VLMs: (1) *Cumulative.* We expect that with more seen samples, the performance should improve as task-specific knowledge accumulates [40, 57]. However, test-time prompt tuning methods [54, 13] treat each test instance independently, resetting to the original model for each new sample, failing to extract historical knowledge from previous test samples. (2) *Multi-modal.* Effective adaptation of VLMs benefits from leveraging knowledge from both textual and visual modalities [28, 35]. However, previous works only capture domain-specific knowledge from a single modality, adapting CLIP based solely on textual [54, 13] or visual [27] feature refinement.

To this end, we propose Dual Prototype Evolving (DPE), a novel test-time VLM adaptation approach that effectively *accumulates* task-specific knowledge from *multi-modalities*, as illustrated in Figure 1. Unlike previous methods that focus on adapting VLMs from a single modality, we create and evolve two sets of prototypes—textual and visual—progressively capturing more accurate multi-modal representations for target classes during test time. To extract historical knowledge from previous test samples, we update these two sets of prototypes online using cumulative average and priority queue strategies, respectively. We further optimize these multi-modal prototypes by introducing learnable residual parameters for each individual test sample to enhance the zero-shot generalization capability of our model. Specifically, rather than solely relying on the entropy minimization objective [62, 79], our DPE also accounts for the alignment between multi-modal prototypes to ensure consistent multi-modal representations. Notably, our DPE requires only the optimization of multi-modal prototypes in the embedding space during test time, eliminating the need to backpropagate gradients through the textual encoder of CLIP, as required in TPT [54] and DiffTPT [13].

The test-time generalization capabilities of our proposed DPE method are extensively evaluated across 15 diverse recognition datasets in two scenarios: natural distribution shifts and cross-dataset generalization. The experimental results validate the superior performance of our DPE, which achieves an average improvement of 3.55% and 4.30% over the state-of-the-art TPT [54] method in these scenarios. Moreover, our proposed DPE achieves this performance while also exhibiting $5\times$ and over $10\times$ test-time efficiency compared to TPT [54] and DiffTPT [13], respectively.

The contributions of this paper are summarized as follows:

- We propose dual prototype evolving (DPE), a novel test-time adaptation method for VLMs that *progressively* captures more accurate *multi-modal* representations for target classes during test time.

- To promote consistent multi-modal representations, we introduce and optimize learnable residuals for each test sample to align the prototypes across modalities.

- Experimental evaluations demonstrate that our DPE consistently outperforms current state-of-the-art methods across 15 diverse datasets while maintaining competitive computational efficiency.

## 2    Related Work

**Vision-Language Models**. Leveraging vast image-text pairs from the Internet, recent large-scale vision-language models (VLMs), such as CLIP [46] and ALIGN [25], have shown remarkable and transferable visual knowledge through natural language supervision [78, 11, 10]. These VLMs enable a "pre-train, fine-tune" paradigm for performing downstream visual tasks, such as image recognition [46, 17, 36], pbject detection [67, 66], and depth estimation [80, 24, 73].

To effectively transfer VLMs to these downstream tasks, researchers have developed two primary methods for adapting the model with few-shot data: prompt learning methods [85, 84, 28, 51, 86, 5] and adapter-based methods [81, 14, 77, 71, 31]. Specifically, prompt learning methods, such as CoOp [85] and CoCoOp [84], focus on learning input prompts with few-shot supervision from downstream data. On the other hand, adapter-based methods, like Tip-Adapter [81] and TaskRes [71], modify the extracted visual or textual representations directly to enhance model performance. However, these approaches often assume the availability of labeled samples from the target domain, which can limit their effectiveness in real-world scenarios. In this work, we address the challenge of test-time adaptation, where the model is required to adapt solely at test time without access to any training samples or ground-truth labels from the target domain. This setting is crucial for real-world deployment, as it allows for robust performance in novel and unseen environments where labeled data cannot be obtained in advance.

**Test-Time Adaptation**. To effectively transfer a model trained on the source domain to the target domain, test-time adaptation methods [62, 79, 60, 3, 59] aim to adjust the model online using a stream of unlabeled test samples. These methods enable the deployment of well-trained models in various out-of-distribution scenarios, thereby enhancing the applicability and reliability of machine learning models in real-world applications [34, 29, 42]. Researchers have applied test-time adaptation techniques successfully across various machine learning tasks, including semantic segmentation [23, 52, 83], human pose estimation [33, 26], and image super-resolution [53, 8].

Recently, increasing research efforts have focused on adapting large-scale VLMs during test time [38, 56, 1, 72, 82, 76]. As the seminal work, Shu *et al*. [54] firstly propose test-time prompt tuning (TPT), which enforces consistency across different augmented views of each test sample. Building on this approach, several subsequent studies have sought to further enhance TPT. For instance, DiffTPT [54] utilizes diffusion-based augmentations to increase the diversity of augmented views, while C-TPT [69] addresses the rise in calibration error during test time prompt tuning. Unlike these approaches, which treat each test sample independently, TDA [27] establishes positive and negative visual caches during test time, enhancing model performance as more samples are processed. Similarly, recent DMN [82] utilizes a dynamic memory to gather information from historical test data. However, these methods solely adapt the model from a single modality perspective, limiting their effectiveness in capturing task-specific knowledge from out-of-distribution domains. Given this, we design DPE to evolve two sets of prototypes from both textual and visual modalities to progressively capture more accurate multi-modal representations for target classes during test time.

## 3    Method

We introduce Dual Prototype Evolving (DPE) as illustrated in Figure 2, to enhance CLIP's zero-shot generalization capabilities across diverse distributions during test time. Unlike previous methods that focus solely on one modality, we design two sets of prototypes, textual and visual, which are progressively updated using the unlabeled test dataset $\mathcal{D}_{\texttt{test}}$.

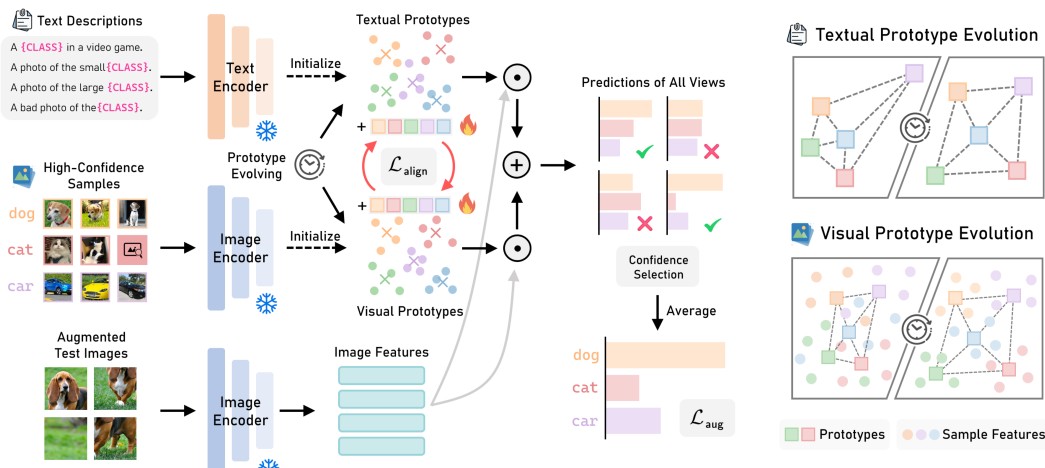

Figure 2: **An overview of our DPE method**. We introduce prototypes from both textual and visual modalities and enable prototype-based inference with CLIP. For each test sample, we optimize both prototypes using learnable residual parameters with alignment loss $\mathcal{L}_{\texttt{align}}$ and self-entropy loss $\mathcal{L}_{\texttt{aug}}$. These prototypes are also progressively evolved over time to capture more accurate and discriminative multi-modal representations for target classes.

## 3.1 Preliminaries

**Zero-Shot CLIP**. CLIP [46] utilizes two pre-trained parallel encoders: a visual encoder $\mathcal{E}_v(\cdot)$ and a textual encoder $\mathcal{E}_t(\cdot)$, which embed images and text descriptions into a shared embedding space $\mathbb{R}^d$. For a $C$-class classification task, CLIP performs zero-shot predictions by computing the similarities between the extracted image feature and the $C$ candidate text features, written as

$$f_v = \mathcal{E}_v(X_{\texttt{test}}), \quad f_{t_c} = \mathcal{E}_t(\mathcal{T}_c), \quad \mathbb{P}_{\texttt{CLIP}}(y = y_c | X_{\texttt{test}}) = \frac{\exp\left(\text{sim}\left(f_{t_c}, f_v\right)/t\right)}{\sum_{t'} \exp\left(\text{sim}\left(f_{t'}, f_v\right)/t\right)}, \quad (1)$$

where $X_{\texttt{test}} \in \mathcal{D}_{\texttt{test}}$ denotes the input test image, and $\mathcal{T}_c$ represents the the class-specific description input for class $y_c$. The pairwise similarities $\text{sim}(\cdot, \cdot)$ are calculated using cosine similarity, and $t$ represents the temperature parameter in the softmax function.

**Test-Time Prompt Tuning**. To enhance the zero-shot generalizability of CLIP, TPT [54] proposes learning an adaptive prompt using the test stream samples. Specifically, for each test sample $X_{\texttt{test}}$, TPT generates $N$ augmented views $\{\mathcal{A}_n(X_{\texttt{test}})\}_{n=1}^N$ and averages the top $\rho$-percentile confident predictions based on an entropy threshold $\tau$ to obtain the final prediction:

$$\mathbb{P}_{\texttt{TPT}}(X_{\texttt{test}}) = \frac{1}{\rho N} \sum_{n=1}^N \mathbb{1}[\mathcal{H}\left(\mathbb{P}_{\texttt{CLIP}}(\mathcal{A}_n(X_{\texttt{test}})) \leq \tau\right] \mathbb{P}_{\texttt{CLIP}}(\mathcal{A}_n(X_{\texttt{test}})). \quad (2)$$

Here, $\mathcal{H}(p) = -\sum_{i=1}^C p_i \log p_i$ calculates the self-entropy of the prediction $p$. The objective of TPT is to optimize the learnable prompt to minimize the self-entropy of the final prediction, *i.e.*, $\min \mathcal{H}(\mathbb{P}_{\texttt{TPT}}(X_{\texttt{test}}))$.

## 3.2 Dual Prototype Evolving

In our DPE method, we construct and iteratively evolve two sets of class-specific prototypes from both visual and textual modalities to achieve a more precise representation of each class over time.

**Textual Prototype Evolution**. In this work, we follow CLIP [46] to use multiple context prompt templates for prompt ensembling. Specifically, for each class $c$, we generate a total of $S$ text descriptions, denoted as $\{\mathcal{T}_c^{(i)}\}_{i=1}^S$. The prototypes of these descriptions in the embedding space are calculated as $\mathbf{t}_c = \frac{1}{S} \sum_i \mathcal{E}_t(\mathcal{T}_c^{(i)})$. To further improve the quality of these prototypes over time, we design them to be updated online through a cumulative average with each individual sample $X_{\texttt{test}}$ in

the test stream. The update rule is given by:

$$\mathbf{t} \leftarrow \frac{(k-1)\mathbf{t} + \mathbf{t}^*}{\|(k-1)\mathbf{t} + \mathbf{t}^*\|}, \quad k \leftarrow k+1, \tag{3}$$

where $\mathbf{t} = [\mathbf{t}_1 \, \mathbf{t}_2 \, \cdots \, \mathbf{t}_C]^\top \in \mathbb{R}^{C \times d}$ is the online updated prototype set, and $\mathbf{t}^* \in \mathbb{R}^{C \times d}$ is the optimized textual prototypes for each individual sample $X_{\texttt{test}}$ in Equation (10). To ensure stable online updates, we set an entropy threshold $\tau_t$ to filter out low-confidence samples (for which $\mathcal{H}(\mathbb{P}_{\texttt{CLIP}}(X_{\texttt{test}})) < \tau_t$) from updating the online prototypes, and maintain a counter $k$ for tracking confident samples.

**Visual Prototype Evolution**. Inspired by TDA [27], we recognize that the historical image features of test images can also be utilized to enhance CLIP's discrimination capability. Therefore, we design a priority queue strategy to store the top-$M$ image features for each class and symmetrically compute a set of visual prototypes that evolve over time. Note that since we cannot access the labels of the test samples, we assign the image features to the queue according to their predicted pseudo-labels. The priority queue for each class $c$ is initialized as empty, denoted as $q_c = \varnothing$. As test samples arrive, we store the image features $f_c$ and the corresponding self-entropy $h_c$ in the priority queue, represented as $q_c = \{(f_c^{(m)}, h_c^{(m)})\}_m$. The elements are sorted by self-entropy $h_c^{(m)}$ such that $h_c^{(m)} < h_c^{(>m)}$. Using this priority queue, the class-specific visual prototype is obtained by: $\mathbf{v}_c = \frac{1}{S_c} \sum_m f_c^{(m)}$, where $S_c \leq M$ denotes the total number of image features stored in the queue.

The priority queues are updated during testing by replacing low-confidence image features with high-confidence ones. Specifically, for each individual test sample $X_{\texttt{test}}$, we first predict the pseudo-label $\ell$ and compute the corresponding self-entropy $h$ as:

$$\ell = \arg\max_{y_c} \mathbb{P}_{\texttt{CLIP}}(y = y_c | X_{\texttt{test}}), \quad h = \mathcal{H}(\mathbb{P}_{\texttt{CLIP}}(X_{\texttt{test}})). \tag{4}$$

Then, we consider the following two scenarios to iteratively update the priority queue $q_\ell$ for class $\ell$: (1) If the priority queue is not full, we directly add the pair $(\mathcal{E}_v(X_{\texttt{test}}), h)$ to the queue; (2) If the priority queue is full and the entropy $h$ of the new sample is lower than the highest entropy value (of the last element) currently in the queue, we replace the highest-entropy element with the new feature and self-entropy $(\mathcal{E}_v(X_{\texttt{test}}), h)$. If $f$ is not lower, we discard the new sample and leave the queue unchanged. After each update, we re-sort the priority queue based on the self-entropy values and re-compute the visual prototypes $\mathbf{v} = [\mathbf{v}_1 \, \mathbf{v}_2 \, \cdots \, \mathbf{v}_C]^\top \in \mathbb{R}^{C \times d}$ for all classes.

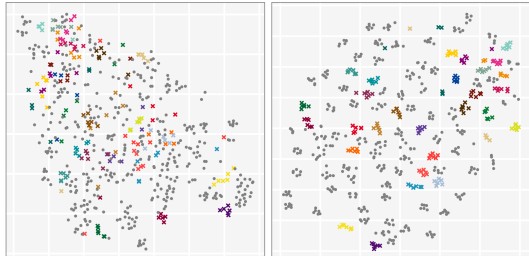

Figure 3: **t-SNE [61] visualizations of the stored image features in the priority queues**. With more samples getting in, the selected image features from each class become more clustered, leading to more representative visual prototypes.

In Figure 3, we present the t-SNE [61] visualizations of the stored image features in the priority queues (with queue size $M = 6$) after updating with 1500 samples (*left*) and 15000 samples (*right*) on the Food101 [2] dataset. We highlight the stored features from 25 random classes using different colors while marking the others in gray. These visualizations illustrate that our priority queue strategy effectively accumulates high-confidence samples, progressively refining the representativeness of the visual prototypes over time.

**Prototype-Based Inference**. Based on our two sets of multi-modal prototypes $\{\mathbf{t}_c\}_{c=1}^C$ and $\{\mathbf{v}_c\}_{c=1}^C$, the final prediction for input $X$ is given by

$$f_v = \mathcal{E}_v(X), \quad \mathbb{P}_{\texttt{Proto}}(y = y_c | X) = \frac{\exp\left(\left(f_v^\top \mathbf{t}_c + \mathcal{A}(f_v^\top \mathbf{v}_c)\right)/t\right)}{\sum_{c'} \exp\left(\left(f_v^\top \mathbf{t}_{c'} + \mathcal{A}(f_v^\top \mathbf{v}_{c'})\right)/t\right)}, \tag{5}$$

Here, $t$ represents the temperature parameter in the softmax function, $\top$ denotes the matrix transpose, and $\mathcal{A}(x) = \alpha \exp\left(-\beta\left(1 - x\right)\right)$ is the affinity function, where $\alpha$ is a balance hyperparameter and $\beta$ is a sharpness ratio. In Appendix A.3, we conduct a sensitivity analysis of these two hyperparameters to evaluate their impact on the overall performance of DPE.

### 3.3 Prototype Residual Learning

To further improve the zero-shot generalizability of our method, we introduce prototype residual learning, which optimizes multi-modal prototypes for each test sample. Unlike previous prompt tuning approaches [54, 13] that require backpropagating gradients through the text encoder to update input prompts, our method directly updates the prototype sets in the embedding space.

Specifically, after being evolved with the last test sample, the dual sets of multi-modal prototypes, denoted as $\mathbf{t} = [\mathbf{t}_1 \, \mathbf{t}_2 \, \cdots \, \mathbf{t}_C]^\top \in \mathbb{R}^{C \times d}$ and $\mathbf{v} = [\mathbf{v}_1 \, \mathbf{v}_2 \, \cdots \, \mathbf{v}_C]^\top \in \mathbb{R}^{C \times d}$, are considered as the initialization for updating with the current test sample. We further introduce learnable residual parameters $\hat{\mathbf{t}} = [\hat{\mathbf{t}}_1 \, \hat{\mathbf{t}}_2 \, \cdots \, \hat{\mathbf{t}}_C]^\top \in \mathbb{R}^{C \times d}$ and $\hat{\mathbf{v}} = [\hat{\mathbf{v}}_1 \, \hat{\mathbf{v}}_2 \, \cdots \, \hat{\mathbf{v}}_C]^\top \in \mathbb{R}^{C \times d}$. These parameters are initialized to zero and are used to optimize the prototypes for each given test input $X_{\texttt{test}}$, denoted as

$$\mathbf{t}_c \leftarrow \frac{\mathbf{t}_c + \hat{\mathbf{t}}_c}{\|\mathbf{t}_c + \hat{\mathbf{t}}_c\|}, \quad \mathbf{v}_c \leftarrow \frac{\mathbf{v}_c + \hat{\mathbf{v}}_c}{\|\mathbf{v}_c + \hat{\mathbf{v}}_c\|}. \tag{6}$$

Similar to Equation (2), we optimize these residual parameters to promote consistent predictions across a total of $N$ different augmented views of the given test image $X_{\texttt{test}}$ using the unsupervised entropy minimization objective:

$$\mathcal{L}_{\texttt{aug}} = \mathcal{H}(\mathbb{P}_{\texttt{DPE}}(X_{\texttt{test}})) = -\sum_{c=1}^{C} \mathbb{P}_{\texttt{DPE}}(y = y_c | X_{\texttt{test}}) \log \mathbb{P}_{\texttt{DPE}}(y = y_c | X_{\texttt{test}}), \tag{7}$$

$$\text{where } \mathbb{P}_{\texttt{DPE}}(X_{\texttt{test}}) = \frac{1}{\rho N} \sum_{n=1}^{N} \mathbb{1}[\mathcal{H}\left(\mathbb{P}_{\texttt{Proto}}(\mathcal{A}_n(X_{\texttt{test}})) \leq \tau\right] \mathbb{P}_{\texttt{Proto}}(\mathcal{A}_n(X_{\texttt{test}})). \tag{8}$$

However, researchers have shown that focusing solely on reducing entropy can lead the model to make overconfident predictions [69]. To address this, we apply an additional constraint to align the multi-modal prototypes during optimization, explicitly enforcing consistent multi-modal representations between dual sets of prototypes. Specifically, we introduce a self-supervised alignment loss that utilizes the contrastive InfoNCE loss [43] to bring prototypes from the same class closer together while pushing prototypes from different classes further apart:

$$\mathcal{L}_{\texttt{align}} = \frac{1}{C} \sum_{c=1}^{C} \left( -\log \frac{\exp(\mathbf{t}_c^\top \mathbf{v}_c)}{\sum_{c'} \exp(\mathbf{t}_c^\top \mathbf{v}_{c'})} - \log \frac{\exp(\mathbf{t}_c^\top \mathbf{v}_c)}{\sum_{c'} \exp(\mathbf{t}_{c'}^\top \mathbf{v}_c)} \right). \tag{9}$$

In summary, the final objective for optimizing the multi-modal prototypes $\mathbf{t}, \mathbf{v}$ is

$$\mathbf{t}^*, \mathbf{v}^* = \arg\min_{\mathbf{t},\mathbf{v}} \left( \mathcal{L}_{\texttt{aug}} + \lambda \mathcal{L}_{\texttt{align}} \right), \tag{10}$$

where $\lambda$ is a scale factor to balance the contribution of the alignment loss. Note that $\mathbf{t}^*$ and $\mathbf{v}^*$ are obtained from a single update step.

After optimizing the prototypes for each test sample, we evolve the online textual prototypes $\mathbf{t}$ as described in Equation (3), and also update the priority queues to re-compute the visual prototypes $\mathbf{v}$. The evolved prototype sets then serve as the initialization for the next test sample, progressively enhancing generalization capability during test-time adaptation.

## 4 Experiments

In this section, we evaluate our proposed method on robustness to natural distribution shifts and cross-datasets generalization across 15 various datasets. Moreover, we also compare the test-time efficiency of our DPE with existing methods. Finally, we provide ablation experiments to systematically analyze the effects of different algorithm components and design choices.

### 4.1 Experimental Settings

**Datasets**. We follow previous work [54, 13] to evaluate our method on two benchmarking scenarios, namely, robustness to natural distribution shifts and cross-datasets generalization. (1) For the

Table 1: **Performance comparisons on robustness to natural distribution shifts**. We present top-1 accuracy (%) results for all evaluated methods employing both ResNet-50 and ViT-B/16 visual backbones of CLIP. The best results are highlighted in **bold**.

| Method | ImageNet | ImageNet-A | ImageNet-V2 | ImageNet-R | ImageNet-S | Average | OOD Average |
|---|---|---|---|---|---|---|---|
| CLIP-ResNet-50 [46] | 58.16 | 21.83 | 51.41 | 56.15 | 33.37 | 44.18 | 40.69 |
| Ensemble | 59.81 | 23.24 | 52.91 | 60.72 | 35.48 | 46.43 | 43.09 |
| CoOp [85] | 63.33 | 23.06 | 55.40 | 56.60 | 34.67 | 46.61 | 42.43 |
| TPT [54] | 60.74 | 26.67 | 54.70 | 59.11 | 35.09 | 47.26 | 43.89 |
| DiffTPT [13] | 60.80 | **31.06** | 55.80 | 58.80 | 37.10 | 48.71 | 45.69 |
| TDA [27] | 61.35 | 30.29 | 55.54 | 62.58 | 38.12 | 49.58 | 46.63 |
| TPS [56] | 61.47 | 30.48 | 54.96 | 62.87 | 37.14 | 49.38 | 46.36 |
| DMN-ZS [82] | **63.87** | 28.57 | 56.12 | 61.44 | 39.84 | 49.97 | 46.49 |
| **DPE (Ours)** | 63.41 | 30.15 | **56.72** | **63.72** | **40.03** | **50.81** | **47.66** |
| CLIP-ViT-B/16 [46] | 66.73 | 47.87 | 60.86 | 73.98 | 46.09 | 59.11 | 57.20 |
| Ensemble | 68.34 | 49.89 | 61.88 | 77.65 | 48.24 | 61.20 | 59.42 |
| CoOp [85] | 71.51 | 49.71 | 64.20 | 75.21 | 47.99 | 61.72 | 59.28 |
| TPT [54] | 68.98 | 54.77 | 63.45 | 77.06 | 47.94 | 62.44 | 60.81 |
| DiffTPT [13] | 70.30 | 55.68 | 65.10 | 75.00 | 46.80 | 62.28 | 60.52 |
| TDA [27] | 69.51 | **60.11** | 64.67 | 80.24 | 50.54 | 65.01 | 63.89 |
| TPS [56] | 70.19 | 60.08 | 64.73 | 80.27 | 49.95 | 65.04 | 63.76 |
| DMN-ZS [82] | **72.25** | 58.28 | 65.17 | 78.55 | **53.20** | 65.49 | 63.80 |
| **DPE (Ours)** | 71.91 | 59.63 | **65.44** | **80.40** | 52.26 | **65.93** | **64.43** |

evaluation of robustness to natural distribution shifts, we assess the performance of our method using the ImageNet [7] dataset alongside its variant out-of-distribution datasets, including ImageNet-A [21], ImageNet-V2 [47], ImageNet-R [19], and ImageNet-Sketch [63]. (2) For cross-datasets generalization tasks, we conduct comprehensive assessments across 10 diverse recognition datasets, including FGVCAircraft [39], Caltech101 [12], StandfordCars [30], DTD [6], EuroSAT [18], Flowers102 [41], Food101 [2], OxfordPets [44], SUN397 [68], and UCF101 [55]. These datasets offer a comprehensive benchmark for evaluating the robustness of various methods across different distributional variations.

**Implementation Details**. We follow previous works [54, 13] to adopt ResNet-50 [16] and ViT-B/16 [9] backbones as the visual encoder of CLIP. In Appendix C.2, we detail the specific hand-crafted prompts utilized for each dataset. Following TPT [54], we generate 63 augmented views for each test image using random resized cropping to create a batch of 64 images. We learn the prototype residual parameters using AdamW [37] optimizer with a learning rate of 0.0005 for a single step. In default, the scale factor $\lambda$ in Equation (10) is set to 0.5, the normalized entropy threshold $\tau_t$ is set to 0.1, and the queue size $M$ is set to 3. For the affinity function in Equation (5), we set $\alpha = 6.0$ and $\beta = 5.0$, respectively. All experiments are conducted on a single 48GB NVIDIA RTX 6000 Ada GPU. To ensure the reliability of our results, we perform each experiment three times using different initialization seeds and report the mean accuracy achieved.

**Baselines**. We compare our method with established test-time adaptation approaches for CLIP: (1) TPT [54], a prompt tuning method that aims to minimize self-entropy across predictions of multiple augmented views; (2) DiffTPT [13], an enhanced version of TPT that utilizes diffusion-based augmentations to optimize prompts; (3) TDA [27], a training-free, adapter-based method which constructs positive and negative caches during test time. (4) TPS [56], an efficient approach that dynamically learns shift vectors for per-class prototypes based solely on the given test sample; (5) DMN-ZS [82], a backpropagation-free method that utilizes a dynamic memory to aggregate information from historical test data. Additionally, we present the zero-shot performance of CLIP using the simple prompt "*a photo of a* {CLASS}" as well as the results from prompt ensembling to show the absolute performance improvements. We also report the performance of CoOp [85], a train-time adaptation method, using 16-shot annotated samples per class on ImageNet. For a fair comparison, we directly report the results of these baselines from their respective original papers. Note that in the DiffTPT [13] paper, the results are based on a subset of the datasets containing 1,000 test samples. This limited sample size may introduce potential imprecision in the reported results.

Table 2: **Performance comparisons on cross-datasets generalization.** We also present top-1 accuracy (%) for all methods on two backbones of CLIP. The best results are highlighted in **bold**.

| Method | Aircraft | Caltech | Cars | DTD | EuroSAT | Flower | Food101 | Pets | SUN397 | UCF101 | Average |
|---|---|---|---|---|---|---|---|---|---|---|---|
| CLIP-ResNet-50 | 15.66 | 85.88 | 55.70 | 40.37 | 23.69 | 61.75 | 73.97 | 83.57 | 58.80 | 58.84 | 55.82 |
| Ensemble | 16.11 | 87.26 | 55.89 | 40.37 | 25.79 | 62.77 | 74.82 | 82.97 | 60.85 | 59.48 | 56.63 |
| CoOp [85] | 15.12 | 86.53 | 55.32 | 37.29 | 26.20 | 61.55 | 75.59 | **87.00** | 58.15 | 59.05 | 56.18 |
| TPT [54] | 17.58 | 87.02 | 58.46 | 40.84 | 28.33 | 62.69 | 74.88 | 84.49 | 61.46 | 60.82 | 57.66 |
| DiffTPT [13] | 17.60 | 86.89 | **60.71** | 40.72 | 41.04 | 63.53 | **79.21** | 83.40 | 62.72 | 62.67 | 59.85 |
| TDA [27] | 17.61 | 89.70 | 57.78 | 43.74 | **42.11** | **68.74** | 77.75 | 86.18 | 62.53 | **64.18** | 61.03 |
| **DPE (Ours)** | 19.80 | 90.83 | 59.26 | **50.18** | 41.67 | 67.60 | 77.83 | 85.97 | **64.23** | 61.98 | **61.93** |
| CLIP-ViT-B/16 | 23.67 | 93.35 | 65.48 | 44.27 | 42.01 | 67.44 | 83.65 | 88.25 | 62.59 | 65.13 | 63.58 |
| Ensemble | 23.22 | 93.55 | 66.11 | 45.04 | 50.42 | 66.99 | 82.86 | 86.92 | 65.63 | 65.16 | 64.59 |
| CoOp [85] | 18.47 | 93.70 | 64.51 | 41.92 | 46.39 | 68.71 | 85.30 | 89.14 | 64.15 | 66.55 | 63.88 |
| TPT [54] | 24.78 | 94.16 | 66.87 | 47.75 | 42.44 | 68.98 | 84.67 | 87.79 | 65.50 | 68.04 | 65.10 |
| DiffTPT [13] | 25.60 | 92.49 | 67.01 | 47.00 | 43.13 | 70.10 | **87.23** | 88.22 | 65.74 | 62.67 | 65.47 |
| TDA [27] | 23.91 | 94.24 | 67.28 | 47.40 | **58.00** | 71.42 | 86.14 | 88.63 | 67.62 | **70.66** | 67.53 |
| **DPE (Ours)** | **28.95** | **94.81** | **67.31** | **54.20** | 55.79 | **75.07** | 86.17 | **91.14** | **70.07** | 70.44 | **69.40** |

## 4.2 Results and Discussions

**Robustness to Natural Distribution Shifts**. In Table 1, we first compare the performance of our method with other state-of-the-art methods on in-domain ImageNet and its 4 out-of-distribution variants. Due to domain shifts, zero-shot CLIP [46] underperforms in out-of-distribution scenarios. As shown in the table, adopting prompt ensembling and prompt learning methods like CoOp [85] can enhance CLIP's generalizability. However, it is important to note that CoOp is a train-time adaptation method that requires an annotated training set, limiting its effectiveness in real-world settings. Despite this, our method still exhibits significant performance gains of 4.20% and 4.21% on average across two different backbones compared to CoOp, indicating the superiority of our DPE in enhancing generalization capability on out-of-distribution domains.

Focusing on test-time adaptation methods, the experimental results demonstrate that our method achieves superior zero-shot generalization performance across various out-of-distribution datasets compared to other approaches. Specifically, our method outperforms existing state-of-the-art prompt tuning methods, surpasses TPT [54] by 3.55% and 3.49% and DiffTPT [13] by 2.10% and 3.65% on average when using ResNet-50 and ViT-B/16 backbones, respectively. Moreover, our method also outperforms cache-based TDA [27] by margins of 1.23% and 0.92% across two different backbones, indicating the effectiveness of our DPE approach. Moreover, our DPE demonstrates performance advantages over the recent TPS [56] and DMN-ZS [82] approaches, outperforming them by 1.43% and 0.84% on average across 5 datasets using the ResNet-50 backbone, further highlighting the superiority of our method. We also demonstrate that our DPE can also be effectively applied to prompts learned using CoOp [85] with a 16-shot ImageNet setup. We compare the performance with other methods on the same 5 datasets in Appendix A.1, where our method consistently demonstrates competitive performance. These results highlight the general effectiveness of our proposed test-time adaptation method in both in-domain and out-of-distribution scenarios.

**Cross-Datasets Generalization**. In Table 2, we further assess the generalizability of our proposed method against other state-of-the-art methods on 10 fine-grained recognition datasets. Given the significant distributional differences, methods may exhibit variable performance across these datasets. Notably, our method, which is not trained on any annotated data, significantly outperforms CoOp [85] by average margins of 5.75% and 5.52% on two respective backbones. When compared to other test-time adaptation methods, DPE exhibits average performance gains of 2.08% to 0.90% compared to DiffTPT and TDA, respectively. On the more advanced ViT-B/16 backbone, DPE continues to outperform existing approaches on 7 out of 10 datasets, with average improvements ranging from 1.87% to 4.30%. These results demonstrate the superior robustness and adaptability of our method in transferring to diverse domains during test time, which is crucial for real-world deployment scenarios.

**Efficiency Comparison**. Table 3 presents a comparison of our method's efficiency against other test-time adaptation approaches for VLMs, evaluated on 50,000 test samples from the ImageNet [7] dataset. The comparison is conducted on a single 48GB NVIDIA RTX 6000 Ada GPU. In our DPE method, the main computational overhead arises from the visual prototype evolution and prototype residual learning components. Specifically, while zero-shot CLIP requires 10.1 ms to infer a single im-

Table 3: **Efficiency comparison on ImageNet [7]**. We report the testing time, the achieved accuracy, and the performance gains compared to zero-shot CLIP.

| Method | Testing Time | Accuracy | Gain |
|---|---|---|---|
| CLIP [46] | 9 min | 59.81 | - |
| TPT [54] | 9 h 15 min | 60.74 | +0.93 |
| DiffTPT [13] | > 20 h | 60.80 | +0.99 |
| TDA [27] | 1 h 5 min | 61.35 | +1.54 |
| TPS [56] | 55 min | 61.47 | +1.66 |
| **DPE (Ours)** | 1 h 50 min | **63.41** | **+3.60** |

age, incorporating our prototype residual learning increases the inference time to 64.7 ms per image. Further including the visual prototype evolution extends this to 132.1 ms per image.

Our proposed method shows improved computational efficiency compared to other prompt tuning methods, for example, $5\times$ faster than TPT [54] and over $10\times$ faster than DiffTPT [13], as it requires only learning the prototype residues without the need to backpropagate gradients through the textual encoder. While our method is less efficient than TDA [27] and TPS [56], as we still backpropagate gradients to update multi-modal prototypes, it offers notable performance advantages.

## 4.3 Ablation Studies

**Different Textual Prototype Evolution Rules**. In Table 4, we report the performance on ImageNet [7] using different textual prototype evolution rules. We have the following key observations: (1) Fully updating our textual prototypes $\mathbf{t}$ to the optimized prototypes $\mathbf{t}^*$ for each individual test image results in collapsed performance; (2) Compared to not evolving the textual prototypes, using an exponential moving average update rule with a decay

Table 4: **Performance comparison using different textual prototype evolution rules on ImageNet [7]**. For each method, we present the update rule formula and report the resulting accuracy on the ImageNet dataset.

| Update Rule | Formula | Accuracy |
|---|---|---|
| No Update | $\mathbf{t} \leftarrow \mathbf{t}$ | 62.93 |
| Full Update | $\mathbf{t} \leftarrow \mathbf{t}^*$ | 21.83 |
| Exponential Avg. | $\mathbf{t} \leftarrow 0.99\mathbf{t} + 0.01\mathbf{t}^*$ | 63.11 |
| Exponential Avg. | $\mathbf{t} \leftarrow 0.95\mathbf{t} + 0.05\mathbf{t}^*$ | 62.57 |
| Cumulative Avg. | $\mathbf{t} \leftarrow ((k-1)\mathbf{t} + \mathbf{t}^*)/k$ | **63.41** |

rate of 0.99 leads to a slight performance improvement of 0.18%; however, setting a lower decay rate of 0.95 decreases the performance by 0.36%. (3) Our cumulative average update rule yields the highest performance, achieving a 0.48% improvement compared to no update on ImageNet [7].

**Hyperparameters for Dual Prototype Evolution**. We provide a sensitivity analysis for the hyperparameters $\tau_t$ and $M$ on the Caltech101 [12] dataset in Figure 4 (*Left*). Specifically, $\tau_t$ represents the normalized entropy threshold for evolving our textual prototypes. When $\tau_t = 0$, our method does not evolve the textual prototypes, leading to a significant performance decrease, as shown in Figure 4 (*Left*). Moreover, setting $\tau_t = 0.1$ results in the highest performance, whereas a higher threshold leads to a slight decrease in performance. Additionally, the queue size $M$ acts as a soft threshold hyperparameter for evolving the visual prototypes. Our setting of $M = 3$ consistently yields the highest performance. Lowering $M$ causes the visual prototypes to fail in capturing the diversity of test samples from the same class, while increasing $M$ introduces additional low-confidence noisy samples that hinder discrimination among target classes. Notably, our DPE method consistently outperforms other approaches across a reasonable range of hyperparameter settings: all combinations of entropy threshold $\tau_t \geq 0.1$ and queue size $M > 3$ achieve over 90.3% accuracy on Caltech101, whereas TPT [54] and TDA [27] only achieve 87.02% and 89.70%, respectively.

**Effects of Different Learnable Modules**. Recall that in our DPE method, we optimize our multi-modal prototypes by introducing two sets of learnable residual parameters $\hat{\mathbf{t}}$ and $\hat{\mathbf{v}}$ for each individual test image. In Figure 4 (*Middle*), we ablate the effects of each set of learnable residual parameters and report the performance across three datasets. Specifically, on ImageNet [7], optimizing only the textual prototypes for individual samples results in a 1.40% improvement, while optimizing only the visual prototypes yields a non-trivial 0.36% improvement, compared to keeping both $\hat{\mathbf{t}}$ and $\hat{\mathbf{v}}$ fixed. Optimizing both sets of residual parameters leads to a further performance increase, *e.g.*, by 1.52% on ImageNet [7]. This indicates both learnable modules contribute to the overall effectiveness of DPE.

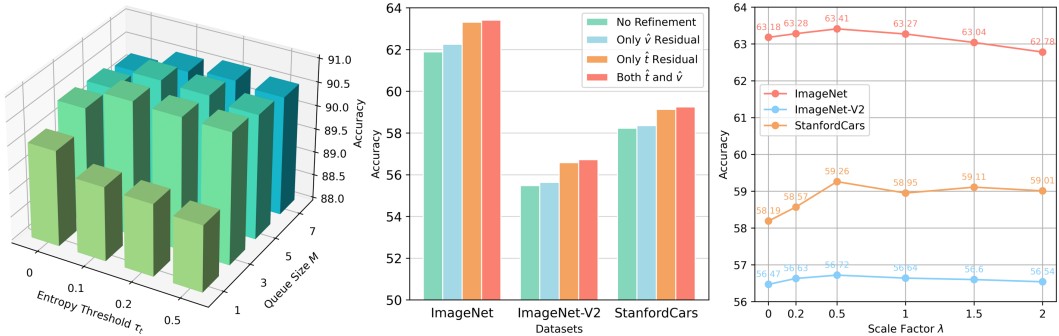

Figure 4: **Ablation studies**. (*Left*) Sensitivity analysis of $\tau_t$ and $M$ on Caltech101 [12]; (*Middle*) Analysis of the performance contributions from various learnable parameter settings across three datasets; (*Right*) Performance on three datasets with varying scale factor $\lambda$ in Equation (10).

**Scaling the Alignment Loss**. Finally, we ablate the effect of the alignment loss by varying the scale factor $\lambda$ in Figure 4 (*Right*). Compared to optimizing solely using entropy minimization loss (*i.e.*, $\lambda = 0$) during test-time adaptation, applying the additional alignment loss results in a performance improvement of 0.23% to 1.07% across three different datasets. However, there is a trade-off between prototype alignment and self-entropy minimization: setting $\lambda$ too high leads to a performance drop. Our experiments show that our setting of $\lambda = 0.5$ yields the highest performance.

**Impact of Varying Update Steps**. In Equation (10), we update the multi-modal prototypes with a single update step for each test instance. To evaluate the impact of different numbers of update steps on overall performance, we conduct ablation experiments by varying the number of update steps from 1 to 5 and report the resulting

Table 5: **Ablation studies on different update steps in prototype residual learning**. We vary the number of update steps from 1 to 5 and report the achieved performance on ImageNet [7].

| # Steps | 1 | 2 | 3 | 4 | 5 |
|---|---|---|---|---|---|
| Accuracy | 63.41 | **63.45** | 63.28 | 63.26 | 63.32 |

performance on ImageNet. As shown in Table 5, the number of update steps does not significantly influence performance (within a range of 0.2%). While increasing the update steps to 2 yields a slight performance gain of 0.04%, it also leads to a proportional decrease in inference efficiency. Given this trade-off, we adopt the single-step update as the default for balancing efficiency and performance.

## 5 Conclusion

In this work, we introduce Dual Prototype Evolving (DPE), a novel and effective approach for enhancing the zero-shot generalizability of VLMs during test time. Unlike previous methods that only focus on adapting the VLMs from one modality, we create and evolve two sets of prototypes—textual and visual—progressively capturing more accurate multi-modal representations for target classes during test time. Moreover, we also introduce prototype residual learning to optimize the dual prototype sets for each individual test sample, which further enhances the test-time generalization capabilities of VLMs. Through comprehensive experiments, we demonstrate that our proposed DPE achieves state-of-the-art performance while also exhibiting competitive test-time efficiency.

**Limitations**. While our proposed DPE method effectively adapts CLIP to out-of-distribution domains during test time, we identify two potential limitations: (1) It still requires gradient backpropagation to optimize the multi-modal prototypes. This optimization process introduces additional computational complexity compared to zero-shot CLIP [46], which may affect its real-time performance in practical deployment scenarios. (2) Since DPE needs to maintain priority queues to evolve the visual prototypes, it increases the memory cost during inference.

**Broader Impacts.** In this work, we aim to build more reliable machine learning systems by leveraging the extensive knowledge of current foundational models, specifically CLIP [46]. Specifically, we follow TPT [54] to apply the test-time adaptation setting to vision-language models to align with real-world scenarios. By employing our DPE approach, the CLIP model can adapt itself to diverse domains during test time, which enhances its practical applicability in real-world deployment scenarios. We hope this work inspires future studies to focus on the generalization and robustness of pre-trained large-scale foundation models.

## Acknowledgements

This work has been funded in part by the Army Research Laboratory (ARL) award W911NF-23-2-0007, DARPA award FA8750-23-2-1015, and ONR award N00014-23-1-2840.

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

# Dual Prototype Evolving for Test-Time Generalization of Vision-Language Models

# Appendix

In this supplementary document, we provide additional details and experimental results to enhance understanding and insights into our method. This supplementary document is organized as follows:

- Full numerical results on robustness to natural distribution shifts are detailed in Section A.1.
- We present additional performance comparisons on larger-scale VLMs, specifically OpenCLIP with a ViT-L/14 backbone, in Section A.2.
- Sensitivity analysis of hyperparameters $\alpha$ and $\beta$ is provided in Section A.3.
- We provide a sensitivity analysis of queue size $M$ on Caltech101 and ImageNet, observing performance trends based on its variations in Section A.4.
- We evaluate the individual impact of each component, showing the significant contribution of VPE, TPE, and PRL to overall performance in Section A.5.
- We analyze the effects of the alignment and self-entropy losses, highlighting their combined performance benefits on ImageNet in Section A.6
- We highlight the differences between our approach and similar methods in Section B.
- Detailed statistics for all utilized datasets are provided in Section C.1.
- We present the specific textual prompts we used for each dataset in Section C.2.
- We list the license information for all used assets in Section D.

## A    Additional Experimental Results

### A.1    Full Results on Robustness to Natural Distribution Shifts

In Table A1, we compare the performance of our method with other state-of-the-art methods on in-domain ImageNet and its 4 out-of-distribution variants. Specifically, we demonstrate that our DPE can also be applied to prompts learned using CoOp [85] with a 16-shot ImageNet setup. Our methods also demonstrates competitive performance compared to other methods. It is also important to notice that, our proposed method accumulates task-specific knowledge over time, therefore can achieve higher performance gain on a larger test set (*e.g.*, ImageNet-R and ImageNet-S).

### A.2    Performance Comparisons on Larger-Scale VLMs

Our DPE method can theoretically be applied to various contrastively pre-trained vision-language models, such as ALIGN [25], LiT [74], and CoCa [70]. In Table A2, we use larger-scale OpenCLIP (ViT-L/14) [4] as an example and compare the performance of TDA and our method on robustness to natural distribution shifts. We can observe that our DPE still outperforms TDA by 1.07% on average across 5 datasets, showcasing that our method generalizes well to larger-scale VLMs.

### A.3    More Sensitivity Analyses of Hyper-Parameters

In our experiments on ImageNet [7], we set the hyperparameters $\alpha$ and $\beta$ as defined in Eq. (5) to 6.0 and 5.0, respectively, as detailed in the implementation section. To thoroughly examine the impact of different hyperparameters, we performed a sensitivity analysis by varying each hyperparameter individually and as-

Table A3: **Sensitivity of hyper-parameters**. All the results are reported on ImageNet [7] using ResNet-50 backbone.

| $\alpha$ | 2.5 | 4.0 | 5.0 | **6.0** | 7.5 | 10.0 |
|---|---|---|---|---|---|---|
| | 62.83 | 63.17 | 63.28 | **63.41** | 63.07 | 62.43 |
| $\beta$ | 2.0 | 3.0 | 4.0 | **5.0** | 6.0 | 7.0 |
| | 62.85 | 63.02 | 63.30 | **63.41** | 63.37 | 63.29 |

sessing the performance on ImageNet with a ResNet-50 backbone, as shown in Table A3. The results show that our selected values of $\alpha = 6.0$ and $\beta = 5.0$ provide the best performance.

Table A1: **Performance comparisons on robustness to natural distribution shifts**. We present top-1 accuracy (%) results for all evaluated methods employing both ResNet-50 and ViT-B/16 visual backbones of CLIP. Additionally, we assess the performance using prompts learned by CoOp [85] with 16-shot training data per class on ImageNet [7]. The best results are highlighted in **bold**.

| Method | ImageNet | ImageNet-A | ImageNet-V2 | ImageNet-R | ImageNet-S | Average | OOD Average |
|---|---|---|---|---|---|---|---|
| CLIP-ResNet-50 [46] | 58.16 | 21.83 | 51.41 | 56.15 | 33.37 | 44.18 | 40.69 |
| Ensemble | 59.81 | 23.24 | 52.91 | 60.72 | 35.48 | 46.43 | 43.09 |
| TPT [54] | 60.74 | 26.67 | 54.70 | 59.11 | 35.09 | 47.26 | 43.89 |
| DiffTPT [13] | 60.80 | **31.06** | 55.80 | 58.80 | 37.10 | 48.71 | 45.69 |
| TDA [27] | 61.35 | 30.29 | 55.54 | 62.58 | 38.12 | 49.58 | 46.63 |
| **Ours** | **63.41** | 30.15 | 56.72 | 63.72 | 40.03 | 50.81 | 47.66 |
| | (± 0.23) | (± 0.41) | (± 0.22) | (± 0.20) | (± 0.11) | (± 0.21) | (± 0.22) |
| CoOp [85] | 63.33 | 23.06 | 55.40 | 56.60 | 34.67 | 46.61 | 42.43 |
| TPT + CoOp [54] | 64.73 | 30.32 | 57.83 | 58.99 | 35.86 | 49.55 | 45.75 |
| DiffTPT + CoOp [13] | 64.70 | **32.96** | **61.70** | 58.20 | 36.80 | **50.87** | **47.42** |
| **Ours + CoOp** | **64.86** | 30.08 | 57.96 | **59.78** | 37.80 | 50.10 | 46.41 |
| | (± 0.18) | (± 0.27) | (± 0.31) | (± 0.19) | (± 0.17) | (± 0.22) | (± 0.23) |
| CLIP-ViT-B/16 [46] | 66.73 | 47.87 | 60.86 | 73.98 | 46.09 | 59.11 | 57.20 |
| Ensemble | 68.34 | 49.89 | 61.88 | 77.65 | 48.24 | 61.20 | 59.42 |
| TPT [54] | 68.98 | 54.77 | 63.45 | 77.06 | 47.94 | 62.44 | 60.81 |
| DiffTPT [13] | 70.30 | 55.68 | 65.10 | 75.00 | 46.80 | 62.28 | 60.52 |
| TDA [27] | 69.51 | **60.11** | 64.67 | 80.24 | 50.54 | 65.01 | 63.89 |
| **Ours** | **71.91** | 59.63 | 65.44 | 80.40 | 52.26 | 65.93 | 64.43 |
| | (± 0.09) | (± 0.18) | (± 0.17) | (± 0.24) | (± 0.11) | (± 0.16) | (± 0.18) |
| CoOp [85] | 71.51 | 49.71 | 64.20 | 75.21 | 47.99 | 61.72 | 59.28 |
| TPT + CoOp [54] | 73.61 | 57.95 | **66.83** | 77.27 | 49.29 | 64.99 | 62.83 |
| DiffTPT + CoOp [13] | **75.00** | 58.09 | 66.80 | 73.90 | 49.50 | 64.12 | 61.97 |
| **Ours + CoOp** | 73.67 | **59.43** | 66.38 | 78.49 | 50.78 | 65.75 | 63.77 |
| | (± 0.14) | (± 0.36) | (± 0.32) | (± 0.06) | (± 0.08) | (± 0.23) | (± 0.26) |

Table A2: **Performance comparisons on robustness to natural distribution shifts**. We present top-1 accuracy (%) results for all evaluated methods employing larger-scale ViT-L/14 visual backbones of OpenCLIP [4]. The best results are highlighted in **bold**.

| Method | ImageNet | ImageNet-A | ImageNet-V2 | ImageNet-R | ImageNet-S | Average | OOD Average |
|---|---|---|---|---|---|---|---|
| OpenCLIP (ViT-L/14) [46] | 74.04 | 53.88 | 67.69 | 87.42 | 63.18 | 69.31 | 68.13 |
| TDA [27] | 76.28 | **61.27** | 68.42 | 88.41 | 64.67 | 71.81 | 70.69 |
| **DPE (Ours)** | **77.87** | 61.09 | **70.83** | **89.18** | **66.33** | **73.06** | **71.86** |

## A.4 Ablation Study on Queue Size

In Figure 4 (*Left*), we provide a sensitivity analysis of queue size $M$ on the Caltech101 dataset. We further analyze the impact of hyperparameter $M$ on larger-scale ImageNet in Table A4. Similar to the results on Caltech101, we observe that the performance increases by 0.5% when adjusting $M$ from 1 to 3 but exhibits a slight decrease of 0.2% when further increasing to 7. We speculate that initially increasing the value of $M$ allows our priority queue to collect more diverse features and obtain representative prototypes. However, further increasing it leads to the inclusion of more low-confidence noisy samples, which has adverse effects.

## A.5 Effectiveness of Each Component

We conduct additional ablation experiments to analyze the individual effect of each component in Table A5. In the table, VPE, TPE, and PRL refer to visual prototype evolution, textual prototype evolution, and prototype residual learning, respectively. Note that Experiment #3 is invalid since TPE requires optimized textual prototypes $t^*$ from PRL. As shown, VPE is the most influential component, providing a ∼2% improvement over zero-shot CLIP. The other two components also contribute significantly to the overall performance.

## A.6 Ablation Study on Two Loss Terms

The alignment loss $\mathcal{L}_{\texttt{align}}$ acts from a global perspective by promoting consistent multi-modal prototypes, ensuring that the representations are aligned for all subsequent test samples. The

Table A4: **Ablation studies on different values of $M$ (priority queue size)**.

| Values of $M$ | 1 | 2 | 3 | 4 | 5 | 6 | 7 |
|---|---|---|---|---|---|---|---|
| ImageNet Acc. | 62.91 | 63.17 | 63.41 | 63.34 | 63.29 | 63.29 | 63.21 |

Table A5: **Effectiveness of different algorithm components**. VPE, TPE, and PRL refer to visual prototype evolution, textual prototype evolution, and prototype residual learning, respectively.

| # | VPE | TPE | PRL | ImageNet Acc. |
|---|---|---|---|---|
| 1 | ✗ | ✗ | ✗ | 59.81 |
| 2 | ✓ | ✗ | ✗ | 61.83 |
| 3 | ✗ | ✓ | ✗ | - |
| 4 | ✗ | ✗ | ✓ | 61.59 |
| 5 | ✓ | ✓ | ✗ | 61.90 |
| 6 | ✓ | ✗ | ✓ | 62.93 |
| 7 | ✗ | ✓ | ✓ | 62.48 |
| 8 | ✓ | ✓ | ✓ | **63.41** |

Table A6: **Effects of self-entropy loss and alignment loss.**. Specifically, we apply the self-entropy loss ($\mathcal{L}_{aug}$) and alignment loss ($\mathcal{L}_{align}$) individually and in combination, and report the accuracy on ImageNet using the ResNet-50 backbone.

| # | $\mathcal{L}_{aug}$ | $\mathcal{L}_{align}$ | ImageNet Acc. |
|---|---|---|---|
| 1 | ✗ | ✗ | 61.90 |
| 2 | ✓ | ✗ | 63.18 |
| 3 | ✗ | ✓ | 62.46 |
| 4 | ✓ | ✓ | **63.41** |

self-entropy loss $\mathcal{L}_{aug}$, in contrast, greedily targets on improving individual sample predictions by penalizing high-entropy predictions across augmented views. To provide a clearer understanding, we analyze the effects of the two loss terms on ImageNet using the ResNet-50 backbone and report the performance in Table A6. We can observe that while the alignment loss alone improves the performance by 0.56%, the self-entropy loss provides a greater performance gain of 1.28%. Combining both loss terms further enhances performance by an additional 0.23%.

## B   Further Discussions on Related Work

We acknowledge that our DPE method shares some high-level ideas with DMN-ZS, TPS, TaskRes and MaPLe. However, there are some key distinctions. Here, we discuss the differences between our method and these approaches, respectively:

- **DMN [82]**. While DMN(-ZS) also utilizes historical test samples to enhance the test-time generalizability of VLMs, it only updates the visual memory online while keeping the textual features/classifier unchanged. Therefore, we consider DMN similar to TDA, as both methods adapt CLIP only from a uni-modal (visual) perspective. In contrast, our DPE is designed to progressively capture more accurate multi-modal representations on the fly with test samples.

- **TPS [56]**. Similarly, since TPS only updates the textual prototypes during testing, we categorize it with TPT and DiffTPT, which also account only for uni-modal (textual) adaptation. Moreover, TPS has similar limitations to TPT, as discussed in Lines 46-49, where it treats each test instance independently, resetting to the original model for each new sample. In contrast, our DPE can accumulate task-specific knowledge as more test samples are processed.

- **TaskRes [71] and MaPLe [28]**. While our method shares some similarities in method details (*e.g.*, multi-modal prototype residuals), we focus on a completely different test-time adaptation setting. Specifically, TaskRes and MaPLe aim to adapt CLIP using labeled few-shot samples, whereas our proposed DPE approach leverages only the unlabeled target data stream to adapt the model to out-of-distribution domains. Moreover, we innovatively propose textual/visual prototype evolution, which enables our method to progressively capture more accurate multi-modal representations during test time. The two works mentioned above, while effective in learning from few-shot samples, do not incorporate such knowledge accumulation techniques.

## C   Additional Implementation Details

### C.1   Dataset Details

In Table C7, we present the detailed statistics of each dataset we used in our experiments, including the number of classes, the sizes of training, validation and testing sets, and their original tasks.

Table C7: **Detailed statistics of datasets used in experiments**. Note that the last 4 ImageNet variant datasets are designed for evaluation and only contain the test sets.

| Dataset | Classes | Training | Validation | Testing | Task |
|---|---|---|---|---|---|
| Caltech101 [12] | 100 | 4,128 | 1,649 | 2,465 | Object recognition |
| DTD [6] | 47 | 2,820 | 1,128 | 1,692 | Texture recognition |
| EuroSAT [18] | 10 | 13,500 | 5,400 | 8,100 | Satellite image recognition |
| FGVCAircraft [39] | 100 | 3,334 | 3,333 | 3,333 | Fine-grained aircraft recognition |
| Flowers102 [41] | 102 | 4,093 | 1,633 | 2,463 | Fine-grained flowers recognition |
| Food101 [2] | 101 | 50,500 | 20,200 | 30,300 | Fine-grained food recognition |
| ImageNet [7] | 1,000 | 1.28M | - | 50,000 | Object recognition |
| OxfordPets [44] | 37 | 2,944 | 736 | 3,669 | Fine-grained pets recognition |
| StanfordCars [30] | 196 | 6,509 | 1,635 | 8,041 | Fine-grained car recognition |
| SUN397 [68] | 397 | 15,880 | 3,970 | 19,850 | Scene recognition |
| UCF101 [55] | 101 | 7,639 | 1,898 | 3,783 | Action recognition |
| ImageNet-V2 [47] | 1,000 | - | - | 10,000 | Robustness of collocation |
| ImageNet-Sketch [63] | 1,000 | - | - | 50,889 | Robustness of sketch domain |
| ImageNet-A [21] | 200 | - | - | 7,500 | Robustness of adversarial attack |
| ImageNet-R [19] | 200 | - | - | 30,000 | Robustness of multi-domains |

Table C8: **Textual prompts used in experiments**. In addition to these prompts, we also employ CuPL [45] prompts to further enhance performance.

| Dataset | Prompts |
|---|---|
| ImageNet [7]
ImageNet-V2 [47]
ImageNet-Sketch [63]
ImageNet-A [21]
ImageNet-R [19] | "itap of a {CLASS}."
"a bad photo of the {CLASS}."
"a origami {CLASS}."
"a photo of the large {CLASS}."
"a {CLASS} in a video game."
"art of the {CLASS}."
"a photo of the small {CLASS}." |
| Caltech101 [12] | "a photo of a {CLASS}." |
| DTD [6] | "{CLASS} texture." |
| EuroSAT [18] | "a centered satellite photo of {CLASS}." |
| FGVCAircraft [39] | "a photo of a {CLASS}, a type of aircraft." |
| Flowers102 [41] | "a photo of a {CLASS}, a type of flower." |
| Food101 [2] | "a photo of {CLASS}, a type of food." |
| OxfordPets [44] | "a photo of a {CLASS}, a type of pet." |
| StanfordCars [30] | "a photo of a {CLASS}." |
| SUN397 [68] | "a photo of a {CLASS}." |
| UCF101 [55] | "a photo of a person doing {CLASS}." |

## C.2 Textual Prompts Used in Experiments

In Table C8, we detail the specific hand-crafted prompts utilized for each dataset.

## D License Information

**Datasets**. We list the known license information for the datasets below:

- MIT License: ImageNet-A [21], ImageNet-V2 [47], ImageNet-R [19], and ImageNet-Sketch [63].
- CC BY-SA 4.0 License: OxfordPets [44].
- Research purposes only: ImageNet [7], StandfordCars [30], DTD [6], FGVCAircraft [39], SUN397 [68].

**Code**. In this work, we also use some code implementations from existing codebase: CLIP [46], CoOp [85], TPT [54], and TDA [27]. The code used in this paper are all under the MIT License.

