# OpenReview forum: "Dual Prototype Evolving for Test-Time Generalization of Vision-Language Models"
_NeurIPS.cc/2024/Conference — NeurIPS 2024 poster_

### Official Review · Reviewer_vqJd · 2024-07-11

**Soundness:** 3
**Presentation:** 4
**Contribution:** 3
**Rating:** 6
**Confidence:** 5

**Summary:**

This paper proposes a novel method for test-time generalization of vision-language models. Specifically, the paper updates two prototypes, textual and visual prompts, online using test samples. Additionally, the authors learn task residuals by aligning the two different modality prototypes, further improving performance. Experiments are conducted on standard benchmarks.

**Strengths:**

1. The paper writing is good.
2. The motivation is clear, and the method design is reasonable.
3. Experiments validate the effectiveness of the method.

**Weaknesses:**

1. More ablation studies are expected: What are the different impacts of the two loss terms in Eq 10?
2. Clarification is needed: Are t and v in Eq 10 achieved after loss convergence, or are they obtained from a single update step? Is there a significant difference between the two?
3. Discussion and comparison with a closely related method (DMN [a]) are expected. Both methods maintain the online queue/memory and weight cached features to obtain new prototypes. The difference is that this paper averages cached features to obtain prototypes, while DMN uses an attention mechanism. The authors should discuss both methods and the differences in prototype generation.

[a] Dual Memory Networks: A Versatile Adaptation Approach for Vision-Language Models, CVPR2024

**Questions:**

See weaknesses.

---

> ### Author Rebuttal · Authors · 2024-08-06
>
> Dear Reviewer vqJd,
>
> We greatly appreciate your valuable feedback on our paper. We address the raised concerns and questions below.
>
> ---
>
> **Comment (1)**: “*More ablation studies are expected: What are the different impacts of the two loss terms in Eq 10?*”
>
> **Response (1)**: Thank you for your valuable feedback. The alignment loss $\mathcal{L}\_{\mathsf{align}}$ acts from a global perspective by promoting consistent multi-modal prototypes, ensuring that the representations are aligned for all subsequent test samples. The self-entropy loss $\mathcal{L}_{\mathsf{aug}}$, in contrast, greedily targets on improving individual sample predictions by penalizing high-entropy predictions across augmented views.
>
> To provide a clearer understanding, we analyze the effects of the two loss terms on ImageNet using the ResNet-50 backbone and report the performance in the following table:
>
>
> |  #   | $\mathcal{L}_{\mathsf{aug}}$ | $\mathcal{L}_{\mathsf{align}}$ | ImageNet Acc. |
> | :--: | :--------------------------: | :----------------------------: | :-----------: |
> |  1   |           &#10008;           |            &#10008;            |     61.90     |
> |  2   |           &#10004;           |            &#10008;            |     63.18     |
> |  3   |           &#10008;           |            &#10004;            |     62.46     |
> |  4   |           &#10004;           |            &#10004;            |     63.41     |
>
> We can observe that while the alignment loss alone improves the performance by 0.56%, the self-entropy loss provides a greater performance gain of 1.28%. Combining both loss terms further enhances performance by an additional 0.23%.
>
> ---
>
> **Comment (2)**: “*Clarification is needed: Are t and v in Eq 10 achieved after loss convergence, or are they obtained from a single update step? Is there a significant difference between the two?*”
>
> **Response (2)**: Thank you for your careful review of our paper and sorry for any confusion caused. In Eq. 10, t* and v* are obtained from a single update step. Following your comments, we have conducted a new ablation experiment on ImageNet using the ResNet-50 visual backbone, varying the number of update steps from 1 to 5. The results are as follows:
>
>
> | Number of Update Steps | 1     | 2     | 3     | 4     | 5     |
> | ---------------------- | ----- | ----- | ----- | ----- | ----- |
> | ImageNet Acc.          | 63.41 | **63.45** | 63.28 | 63.26 | 63.32 |
>
>
> As shown, the number of update steps does not significantly influence performance (within a range of 0.2%). Although setting the update steps to 2 slightly increases performance, it also linearly decreases inference efficiency. Therefore, we use a single-step update by default.
>
> We have specified this setting in revised Section 3.3, and will include this ablation experiment in the revised manuscript.
>
> ---
>
> **Comment (3)**: “*Discussion and comparison with a closely related method (DMN [a]) are expected. The authors should discuss both methods and the differences in prototype generation*.”
>
> **Response (3)**: Thank you for pointing out this. We will respond from three perspectives:
>
> - **Motivation**: While DMN(-ZS) also utilizes historical test samples to enhance the test-time generalizability of VLMs, it is important to note that DMN only updates the visual memory online while keeping the textual features/classifier unchanged. Therefore, we consider DMN similar to TDA, as both methods adapt CLIP only from a uni-modal (visual) perspective. In contrast, as we motivated in Section 1, our DPE is designed to progressively capture more accurate multi-modal representations on the fly with test samples.
>
> - **Technical Details**: Focusing solely on the visual modality, the method formulation also differs. DMN-ZS constructs a large dynamic memory (e.g., 50 image features per class) and, during testing, computes the similarity with all features in the memory for each test sample to obtain attention weights. In contrast, our DPE evolves a single representation (i.e., prototype) for each class by maintaining a relatively very small priority queue (e.g., M=3 features per class). Our prototype-based inference requires evaluating only the similarity between test features and prototype features. This technical difference results in our DPE requiring 3x less GPU memory on ImageNet compared to DMN-ZS, as shown in the following table.
>
>
>     | Method     | DMN-ZS   | Ours    | Ours (w/o priority queue) |
>     | ---------- | -------- | ------- | ------------------------- |
>     | GPU Memory | 14110 MB | 4474 MB | 2138 MB                   |
>
> - **Performance Comparison**: We have also included a performance comparison between DPE and DMN-ZS on robustness to natural distribution shifts using the ResNet-50 backbone of CLIP. The results are as follows:
>
>
>     | Method         | ImageNet  | ImageNet-A | ImageNet-V2 | ImageNet-R | ImageNet-S |  Average  | OOD Average |
>     | -------------- | :-------: | :--------: | :---------: | :--------: | :--------: | :-------: | :---------: |
>     | DMN-ZS         | **63.87** |   28.57    |    56.12    |   61.44    |   39.84    |   49.97   |    46.49    |
>     | **DPE (Ours)** |   63.41   | **30.15**  |  **56.72**  | **63.72**  | **40.03**  | **50.81** |  **47.66**  |
>
>
>     As shown in the table, our proposed DPE outperforms DMN-ZS by 0.84% on average across 5 datasets, demonstrating the superiority of our method. Additionally, our method also outperforms DMN-ZS on the ViT-B/16 backbone. We have included the full results in Table 1 and added the corresponding discussions about DMN in the revised paper.
>
> ---
>
> We hope that our responses have addressed your concerns. If you have additional comments or concerns, please let us know and we will be more than happy to answer.
>
>
> Best,
>
> Authors

---

> > ### Comment · Reviewer_vqJd · 2024-08-12
> >
> > Thanks for your response. My concern has been addressed. I will increase my score to 6.

---

> > > ### Author Response · Authors · 2024-08-12
> > >
> > > Dear Reviewer vqJd,
> > >
> > > Thank you for your insightful review of our paper. We greatly appreciate your positive recommendation!
> > >
> > > Best,
> > >
> > > Authors

---

### Official Review · Reviewer_SVvN · 2024-07-12

**Soundness:** 3
**Presentation:** 3
**Contribution:** 3
**Rating:** 7
**Confidence:** 4

**Summary:**

The paper introduces a novel test-time adaptation approach for vision-language models (VLMs) called Dual Prototype Evolving (DPE). The method effectively accumulates task-specific knowledge from multi-modalities by creating and evolving two sets of prototypes—textual and visual—during test time. This approach ensures more accurate multi-modal representations for target classes and promotes consistent representations by optimizing learnable residuals. Extensive experiments on 15 benchmark datasets demonstrate that DPE consistently outperforms previous state-of-the-art methods in both performance and computational efficiency.

**Strengths:**

+Originality: The introduction of dual prototypes (textual and visual) for evolving task-specific knowledge at test time is a novel concept that significantly improves the generalization capabilities of VLMs.
+Quality: The experimental results are comprehensive and well-documented, covering a wide range of datasets and scenarios to validate the effectiveness of the proposed method.
+Clarity: The paper is well-written and clearly explains the methodology, including detailed descriptions of the prototypes' evolution and optimization processes.
+Significance: The proposed method addresses a crucial challenge in real-world applications by enabling efficient and effective test-time adaptation without the need for annotated samples from the target domain.

**Weaknesses:**

-Complexity: The introduction of learnable residuals and the dual prototype evolution mechanism adds complexity to the model, which might pose implementation challenges for practitioners.
-Computational Cost: While the paper claims competitive computational efficiency, the requirement to maintain and update priority queues for visual prototypes can increase memory and computational costs, particularly for large-scale datasets.
-Generalization to Other VLMs: The paper primarily focuses on CLIP, and it is unclear how well the proposed method generalizes to other vision-language models or tasks beyond those evaluated.

**Questions:**

1. Can you provide more details on the computational overhead introduced by the dual prototype evolution mechanism compared to baseline methods?
2. How does the proposed method perform when applied to vision-language models other than CLIP? Have you conducted any preliminary experiments in this direction?
3. Could you elaborate on the potential limitations of the prototype residual learning approach and how it might affect the model's performance in different scenarios?

**Limitations:**

The authors have adequately addressed the limitations and potential negative societal impacts of their work. The paper mentions two primary limitations: the additional computational complexity introduced by gradient back-propagation for optimizing multi-modal prototypes and the increased memory cost due to maintaining priority queues. The authors provide constructive suggestions for future work to address these issues, emphasizing the need for more efficient optimization techniques and memory management strategies.

---

> ### Author Rebuttal · Authors · 2024-08-06
>
> Dear Reviewer SVvN,
>
> Thank you for your insightful comments and positive recommendation of our work. We provide point-by-point responses to address your concerns below.
>
> ---
>
> **Comment (1)**: “*The introduction of learnable residuals and the dual prototype evolution mechanism adds complexity to the model, which might pose implementation challenges for practitioners*.”
>
> **Response (1)**: While it is true that the introduction of learnable residuals and the dual prototype evolution mechanism adds complexity compared to zero-shot CLIP, we believe the idea of this work is straightforward and core components of our method can be implemented in under 100 lines of code. Besides, as we presented in Section 4, our DPE method significantly enhances the test-time generalization capabilities of the CLIP model. Therefore, we believe that this added complexity is worthwhile. Please also be assured that we will make the source code publicly available and provide detailed instructions upon acceptance to facilitate easier reproduction.
>
> ---
>
> **Comment (2)**: “*Can you provide more details on the computational overhead introduced by the dual prototype evolution mechanism compared to baseline methods?*”
>
> **Response (2)**: Thank you for your insightful feedback! Here, we provide additional details regarding our computational overhead, including both inference time and memory usage.
>
> - **Inference time**: In our DPE method, the major computational time comes from the visual prototype evolution and prototype residual learning components. Specifically, while zero-shot CLIP requires 10.1 ms to infer one image, including our prototype residual learning increases the inference time to 64.7 ms per image. Further including the visual prototype evolution extends this to 132.1 ms per image. As a reference, TPT requires 666.3 ms per image, while TDA is more efficient, using 73.5 ms per image.
>
> - **Memory**: As you mentioned, maintaining priority queues does indeed increase memory usage. Following your comments, we compared the GPU memory usage on large-scale ImageNet with and without the priority queue mechanism. Specifically, while our DPE method without the priority queue takes 2136 MB of GPU memory, including the priority queue mechanism takes 4474 MB, which doubles the GPU consumption. However, our method still shows a memory advantage compared to TPT, which takes 18701 MB to perform inference on ImageNet.
>
> ---
>
> **Comment (3)**: “*How does the proposed method perform when applied to vision-language models other than CLIP? Have you conducted any preliminary experiments in this direction?*”
>
> **Response (3)**: While our DPE method can theoretically be applied to various contrastively pre-trained vision-language models, such as ALIGN [ICML ’21], LiT [CVPR ’22], and CoCa [TMLR ’22], most of these methods are closed-source, preventing us from evaluating the effectiveness of our method on these models. Our method can certainly be applied to open-source CLIP-style VLMs, such as OpenCLIP [CVPR ’23], MetaCLIP [ICLR ’24], SigLIP [ICCV ’23], and DFN [ICLR ’24]. Here, we use larger-scale OpenCLIP (ViT-L/14) as an example and compare the performance of TDA and our method on robustness to natural distribution shifts:
>
>
>
> | Method              | ImageNet  | ImageNet-A | ImageNet-V2 | ImageNet-R | ImageNet-S | Average   | OOD Average |
> | ------------------- | --------- | ---------- | ----------- | ---------- | ---------- | --------- | ----------- |
> | OpenCLIP (ViT-L/14) | 74.04     | 53.88      | 67.69       | 87.42      | 63.18      | 69.31     | 68.13       |
> | TDA*                | 76.28     | **61.27**  | 68.42       | 88.41      | 64.67      | 71.81     | 70.69       |
> | **DPE (Ours)**      | **77.87** | 61.09      | **70.83**   | **89.18**  | **66.33**  | **73.06** | **71.86**   |
>
> $^*$ We evaluate TDA using the codes provided by the authors.
>
> We can observe that our DPE still outperforms TDA by 1.07% on average across 5 datasets, showcasing that our method generalizes well to larger-scale VLMs.
>
> ---
>
> **Comment (4)**: “*Could you elaborate on the potential limitations of the prototype residual learning approach and how it might affect the model's performance in different scenarios?*”
>
> **Response (4)**: Thank you for your insightful comments. As we discussed in our **Response (2)**, our prototype residual learning requires backpropagation, which increases the inference time from 10.1 ms per image in zero-shot CLIP to 64.7 ms per image. However, as shown in Figure 4 (Middle), our prototype residual learning also significantly enhances the zero-shot generalizability of the CLIP model. This represents a trade-off between efficiency and accuracy: in some real-time scenarios (e.g., decision-making in autonomous driving), simpler and more efficient methods like zero-shot CLIP may be a better fit. In contrast, for scenarios requiring higher precision (e.g., medical image analysis), our method can provide enhanced zero-shot robustness to the CLIP model.
>
> That said, we believe the DPE method strikes a great balance by achieving state-of-the-art performance (+3.5% compared to TPT) with competitive computational efficiency (5x faster than TPT), which holds the potential to inspire future research.
>
> ---
>
> We hope that our responses have addressed your concerns. If you have additional comments or concerns, please let us know and we will be more than happy to answer.
>
>
> Best,
>
> Authors

---

> ### Comment · Reviewer_SVvN · 2024-08-08
>
> Thank you for thoroughly addressing my concerns. After reviewing the comments and responses for the other reviewers, I see that their concerns have also been resolved. The authors have provided clear definitions of terms for better understanding, conducted additional experiments to further evaluate the effectiveness of the proposed method, and offered more in-depth analysis of how the proposed method works in various settings.
>
> Overall, the rebuttal enhances my confidence in this paper. With careful consideration, I believe this paper with revision is worthy of NeurIPS and will significantly impact the test-time generalization field. My final decision is “accept.”

---

> > ### Author Response · Authors · 2024-08-08
> >
> > Dear Reviewer SVvN,
> >
> > We sincerely appreciate the time and effort you invested in reviewing our manuscript.  We greatly appreciate your positive recommendation!
> >
> > Best,
> >
> > Authors

---

### Official Review · Reviewer_owvr · 2024-07-23

**Soundness:** 3
**Presentation:** 3
**Contribution:** 2
**Rating:** 5
**Confidence:** 4

**Summary:**

The paper proposes a novel test-time adaptation method for CLIP models, drawing inspiration from previous works on prototype learning and CLIP-based adaptors.

For each test sample, both textual and visual prototypes are optimized using learnable residual parameters based on alignment loss and self-entropy loss. These prototypes are progressively updated to better utilize the stream of test samples.

Experimental results on common benchmarks demonstrate the proposed method's effectiveness over recent baselines. Additionally, the authors provide comprehensive ablation studies to highlight the impact of hyperparameters.

**Strengths:**

The method is a novel combination of existing methods.

The experiments and ablation studies are comprehensive.

The paper is mostly well written and the structure is clear.

**Weaknesses:**

- The method combines multiple existing techniques, each of which has been explored in previous works. For instance, learnable residuals have been studied in [1], where task residuals were used for CLIP-based adaptation, and learnable multi-modal representations were used in [2] with a similar motivation for CLIP-based adaptation.

- The performance gain appears marginal  (Table 1), especially given the extensive hyperparameter tuning required. The results are sensitive to hyperparameters (Section 4.3). This may raise concerns about whether the method's efficacy is inherent or merely a result of meticulous hyperparameter tuning. Note that influential factors in the proposed method include (1) the entropy threshold, (2) queue size, (3) weight scale factor, and (4) prototype update rules.



- There seems to be an overstatement of the method's applicability. While the title and introduction suggest broad applicability to "Vision-Language Models," the method is only tested on CLIP models, with no experiments conducted beyond CLIP.

[1] Yu et al., Task Residual for Tuning Vision-Language Models, CVPR 2023.

[2] Khattak et al., MaPLe: Multi-modal Prompt Learning, CVPR 2023.

**Questions:**

- Is the proposed method applicable to non-CLIP models?

- Can a simplified version of the method be presented by removing certain components? Specifically, which components are the most influential? From the ablation studies, it appears that each component functions as an add-on.

**Limitations:**

Yes

---

> ### Author Rebuttal · Authors · 2024-08-06
>
> Dear Reviewer owvr,
>
> Thanks for your valuable feedback!
>
> ---
>
> **Comment (1)**: “*The method combines multiple existing techniques …*”
>
> **Response (1)**: While our method shares some similarities in method details (e.g., multi-modal prototype residuals), we focus on a completely different test-time adaptation setting. Specifically, TaskRes [1] and MaPLe [2] aim to adapt CLIP using **labeled** few-shot samples, whereas our proposed DPE approach leverages only the **unlabeled** target data stream to adapt the model to out-of-distribution domains.
>
> Moreover, we innovatively propose textual/visual prototype evolution, which enables our method to *progressively* capture more accurate multi-modal representations during test time. The two works mentioned above, while effective in learning from few-shot samples, do not incorporate such knowledge accumulation techniques. In our **Comment (4)**, we demonstrate that textual/visual prototype evolution contributes significantly to the overall effectiveness of our method.
>
> ---
>
> **Comment (2)**: “*The performance gain appears marginal (Table 1), especially given the extensive hyperparameter tuning required*.”
>
> **Response (2)**: Thanks for your thoughtful feedback. We will respond in two aspects.
>
> - **Our Performance**: While our performance gain may seem modest at first glance, it is important to note that the test-time adaptation setting for CLIP is inherently challenging. Consequently, performance improvements across methods on this benchmark have been limited over the past few years:
>
>
>     | Method      |   CLIP   |   TPT   |  SwapPrompt   |    DiffTPT  |      TDA    |
>     | --------------- | :------: | :-----------: | :-----------: | :-----------: | :-----------: |
>     | Venue       | ICML '21 |  NeurIPS '22  |  NeurIPS '23  |   ICCV '23   |  CVPR '24  |
>     | Accuracy (Gain) |  46.61   | 47.26 (+0.65) | 47.86 (+0.50) | 48.71 (+0.85) | 49.58 (+0.87) |
>
>     Therefore, we believe our performance gain of 1.23% compared to TDA in such a challenging task is not trivial, and holds the potential to inspire future research. Moreover, we performed a t-test and found a p-value of 0.0036, demonstrating that our method's performance is significantly better than TDA.
>
> - **Hyperparameters**: We acknowledge that our method involves several hyperparameters that influence its efficacy. However, our DPE method consistently outperforms other approaches across a reasonable range of hyperparameter settings. For instance, as shown in Fig. 4 (Left), all combinations of entropy threshold $\tau_t \geq 0.1$ and queue size $M \geq 3$ achieve >90.3% accuracy on Caltech101, whereas TPT and TDA only achieve 87.02% and 89.70%, respectively.
>
> ---
>
> **Comment (3)**: “*Is the proposed method applicable to non-CLIP models?*”
>
> **Response (3)**: Thank you for pointing out this. While our DPE method can theoretically be applied to various contrastively pre-trained vision-language models, such as ALIGN, LiT, and CoCa, most of these methods are closed-source, preventing us from evaluating the effectiveness of our method on these models. Our method can certainly be applied to open-source CLIP-style VLMs, such as OpenCLIP [CVPR ’23], MetaCLIP [ICLR ’24], and DFN [ICLR ’24]. Here, we use OpenCLIP (ViT-L/14) as an example to compare our method with TDA:
>
>
> | Method              | ImageNet  | ImageNet-A | ImageNet-V2 | ImageNet-R | ImageNet-S | Average   | OOD Average |
> | ------------------- | --------- | ---------- | ----------- | ---------- | ---------- | --------- | ----------- |
> | OpenCLIP (ViT-L/14) | 74.04     | 53.88      | 67.69       | 87.42      | 63.18      | 69.31     | 68.13       |
> | TDA               | 76.28     | **61.27**  | 68.42       | 88.41      | 64.67      | 71.81     | 70.69       |
> | **DPE (Ours)**      | **77.87** | 61.09      | **70.83**   | **89.18**  | **66.33**  | **73.06** | **71.86**   |
>
> We can see that our DPE still outperforms TDA by 1.25% on average across 5 datasets, showcasing that our method generalizes well to larger-scale VLMs.
>
> Besides, in this work, we followed common practices [1-3] in this field by using the CLIP model as a representative VLM due to its simplicity in design and wide applicability, without loss of generality [3]. We have specified the scope to CLIP in the abstract and introduction sections of the revised manuscript.
>
> [1] Learning to Prompt for Vision-Language Models, IJCV 2022.
>
> [2] Task Residual for Tuning Vision-Language Models, CVPR 2023.
>
> [3] Test-Time Prompt Tuning for Zero-Shot Generalization in Vision-Language Models, NeurIPS 2022.
>
> ---
>
> **Comment (4)**: “*Can a simplified version of the method be presented by removing certain components? Specifically, which components are the most influential?*”
>
> **Response (4)**: Thanks for your insightful question. Following your comments, we conducted additional ablation experiments to analyze the individual effect of each component:
>
> |  #   |   VPE    |   TPE    |   PRL    | ImageNet Acc. |
> | :--: | :------: | :------: | :------: | :-----------: |
> |  1   | &#10008; | &#10008; | &#10008; |     59.81     |
> |  2   | &#10004; | &#10008; | &#10008; |     61.83     |
> |  3   | &#10008; | &#10004; | &#10008; |       -       |
> |  4   | &#10008; | &#10008; | &#10004; |     61.59     |
> |  5   | &#10004; | &#10004; | &#10008; |     61.90     |
> |  6   | &#10004; | &#10008; | &#10004; |     62.93     |
> |  7   | &#10008; | &#10004; | &#10004; |     62.48     |
> |  8   | &#10004; | &#10004; | &#10004; |     63.41     |
>
> VPE, TPE, and PRL refer to visual prototype evolution, textual prototype evolution, and prototype residual learning, respectively. Note that Experiment #3 is invalid since TPE requires optimized textual prototypes t* from PRL. As shown, VPE is the most influential component, providing a ~2% improvement over zero-shot CLIP. The other two components also contribute significantly to the overall performance.
>
> ---
>
> If you have additional comments or concerns, please let us know.
>
> Best,
>
> Authors

---

> > ### Comment · Reviewer_owvr · 2024-08-13
> >
> > I appreciate the authors for the detailed response and additional experiments. It would be great to include some of these results in the main paper. My primary concern remains the marginal improvement of the proposed approach given its complexity. Despite the performance, I still lean towards recommending acceptance of the paper for its technical contribution.

---

> ### Author Response · Authors · 2024-08-13
> **Thanks for your positive recommendation**
>
> Dear Reviewer owvr,
>
> Thank you for your encouraging positive recommendation. We greatly appreciate your recognition of the technical contributions of our work. We will incorporate these additional experiments in our revision carefully. Thank you again for your valuable feedback on our paper.
>
> Best,
>
> Authors

---

### Official Review · Reviewer_Sqqz · 2024-07-26

**Soundness:** 3
**Presentation:** 3
**Contribution:** 3
**Rating:** 5
**Confidence:** 5

**Summary:**

1. This paper proposes a novel test-time adaptation method (DPE) for VLMs that captures multi-modal representations for target classes during test time.
2. This paper introduces and optimizes learnable residuals for each test sample to align the prototypes across modalities.
3. The results of this paper are promising while maintaining competitive computational efficiency.

**Strengths:**

1. The idea is very clear and easy to follow. This paper proposes that former methods just focus on the single-modality, and the proposed method DPE can learn from two-modalities. Meanwhile, DPE accumulates task-specific knowledge in a residual manner, which maintains computational efficiency.
2. The experimental results are promising. Compared with other related methods, DPE achieves improvement on 15 benchmark datasets, and DPE shows improved computational efficiency compared with TPT and DiffTPT, which is practical for the test-time scenarios.
3. The writing and presentation of this paper is good.

**Weaknesses:**

1. The core idea is not novel enough. Many concurrent works with TDA which is DPE's main comparison method have shown a similar idea, such as DMN-ZS[1] actually utilizes the information of two modalities while not demanding any backpropagation, and TPS[2] which has been cited in the paper proposes that residual prototype is useful for test-time prompt learning. Though the specific designs of DPE are different from them, the core idea is somewhat similar.
2. More analysis of hyperparameters is needed. Though the introduction of visual information is helpful intuitively, hyperparameters such as (top-)M may affect the performance obviously, which should be explored comprehensively.
3. More analysis of loss functions is needed. Alignment loss is helpful for performance improvement, but too much alignment leads to a performance drop, the reason should be discussed.
4. Efficiency comparison is not comprehensive. Though DPE performs faster than TPT and DiffTPT, the efficiency comparison with backpropagation-free methods (such as TDA,TPS, and DMN) should be provided.

[1] Dual Memory Networks: A Versatile Adaptation Approach for Vision-Language Models.
[2] Just shift it: Test-time prototype shifting for zero-shot generalization with vision-language models.

**Questions:**

1. Many concurrent works (such as DMN-ZS, TPS) have shown similar ideas, and comparisons with them including the core idea design and performance should be provided.
2. A detailed analysis of hyperparameter M should be provided, and how to choose the M to obtain robust performance in the test-time scenarios also should be discussed.
3. A detailed analysis of loss functions should be provided. As shown in the paper, more alignment brings performance drop, the reason should be discussed. By the way, should the text prototype be aligned in the direction of the visual prototype? Or does the visual prototype only need to be aligned in the direction of the text? Are there any related experiments?
4. Efficiency comparison with backpropagation-free methods (such as TDA,TPS, andDMN) should be provided.

**Limitations:**

1. The core idea is not novel enough, and it seems to be a combination of existing several ideas.
2. Detailed analysis of hyperparameters and loss functions is missing.
3. Efficiency comparison is not comprehensive.

---

> ### Author Rebuttal · Authors · 2024-08-06
>
> Dear Reviewer Sqqz,
>
> We really appreciate your thorough review of our paper!
>
> ---
>
> **Comment (1)**: “*Comparisons with DMN-ZS [1] and TPS [2] including the core idea design and performance should be provided*.”
>
> **Response (1)**: Thank you for pointing this out. We acknowledge that our DPE method shares some high-level ideas with DMN-ZS and TPS. However, there are some key distinctions. Here, we discuss the differences between our method and these two approaches, respectively:
>
> - **DMN-ZS**: While DMN(-ZS) also utilizes historical test samples to enhance the test-time generalizability of VLMs, it only updates the visual memory online while keeping the textual features/classifier unchanged. Therefore, we consider DMN similar to TDA, as both methods adapt CLIP only from a uni-modal (visual) perspective. In contrast, our DPE is designed to progressively capture more accurate **multi-modal representations** on the fly with test samples.
>
> - **TPS**: Similarly, since TPS only updates the textual prototypes during testing, we categorize it with TPT and DiffTPT, which also account only for uni-modal (textual) adaptation. Moreover, TPS has similar limitations to TPT, as discussed in Lines 46-49, where it treats each test instance independently, resetting to the original model for each new sample. In contrast, our DPE can **accumulate** task-specific knowledge as more test samples are processed.
>
> We have also included a performance comparison with these methods on robustness to natural distribution shifts:
>
>
>
> | Method         | ImageNet  | ImageNet-A | ImageNet-V2 | ImageNet-R | ImageNet-S |  Average  | OOD Average |
> | -------------- | :-------: | :--------: | :---------: | :--------: | :--------: | :-------: | :---------: |
> | CLIP-RN50      |   58.16   |   21.83    |    51.41    |   56.15    |   33.37    |   44.18   |    40.69    |
> | TPS           |   61.47   | **30.48**  |    54.96    |   62.87    |   37.14    |   49.38   |    46.36    |
> | DMN-ZS         | **63.87** |   28.57    |    56.12    |   61.44    |   39.84    |   49.97   |    46.49    |
> | **DPE (Ours)** |   63.41   |   30.15    |  **56.72**  | **63.72**  | **40.03**  | **50.81** |  **47.66**  |
>
> As shown, our proposed DPE outperforms TPS and DMN-ZS by 1.43% and 0.84% on average across 5 datasets, demonstrating the superiority of our method. We have updated the results in Table 1.
>
> ---
>
> **Comment (2)**: “*A detailed analysis of hyperparameter M should be provided, and how to choose the M … should be discussed*.”
>
> **Response (2)**: Thank you for your insightful comments. In Figure 4 (Left), we provided a sensitivity analysis of $M$ on the Caltech101 dataset. Following your comments, we further analyze the impact of hyperparameter $M$ on larger-scale ImageNet:
>
>
> | Values of $M$ | 1     | 2     | 3         | 4     | 5     | 6     | 7     |
> | ------------- | ----- | ----- | --------- | ----- | ----- | ----- | ----- |
> | ImageNet Acc. | 62.91 | 63.17 | **63.41** | 63.34 | 63.29 | 63.28 | 63.21 |
>
>
> Similar to the results on Caltech101, we observe that the performance increases by 0.5% when adjusting $M$ from 1 to 3 but exhibits a slight decrease of 0.2% when further increasing $M$ to 7. We speculate that initially increasing the value of $M$ allows our priority queue to collect more diverse features and obtain representative prototypes. However, further increasing it leads to the inclusion of more low-confidence noisy samples, which has adverse effects.
>
> ---
>
> **Comment (3)**: “*... more alignment brings performance drop, the reason should be discussed. By the way, should the text prototype be aligned in the direction of the visual prototype? Are there any related experiments?*”
>
> **Response (3)**: The alignment loss $\mathcal{L}\_{\mathsf{align}}$ acts from a global perspective by promoting consistent multi-modal prototypes, ensuring that the representations are aligned for all subsequent test samples. The self-entropy loss $\mathcal{L}_{\mathsf{aug}}$, in contrast, greedily targets on improving individual sample predictions by penalizing high-entropy predictions across augmented views. Therefore, overly emphasizing alignment may cause the method to prioritize global consistency over the refinement of individual predictions, leading to less accurate results for specific samples.
>
> Besides, rather than solely aligning the visual prototypes in the direction of the text or vice versa, our DPE mutually aligns the textual and visual prototypes, updating both prototypes simultaneously. In Figure 4 (Middle), we keep either the textual or visual prototypes fixed and only align the prototypes from the other modality towards the fixed prototypes. We demonstrated that optimizing prototypes from both modalities achieves the best performance gain of 1.52%.
>
> ---
>
> **Comment (4)**: “*The efficiency comparison with backpropagation-free methods should be provided*.”
>
> **Response (4)**: Thank you for your insightful feedback. We would like to clarify a detail regarding your comment: TPS actually requires backpropagation with self-entropy loss for each sample, similar to TPT.
>
> Following your comment, we compare the inference time per image for each method (using a single A6000 Ada GPU):
>
>
>
> | Method              | CLIP | TPT   | DiffTPT | TPS  | TDA  | TDA* | DMN-ZS | DMN-ZS* | Ours | Ours* |
> | ------------------- | ---- | ----- | ------- | ---- | ---- | ---- | ------ | ------- | ---- | ----- |
> | Inference time (ms) | 10.1 | 668.3 | >2000   | 65.2 | 13.4 | 73.5 | 11.8   | 61.6    | 66.7 | 132.1 |
>
>
> In the table, $^*$ indicates that pseudo-label predictions are enhanced with $N=64$ augmented views and confidence selection, which increases inference time. As shown, our DPE has a inference speed that is half that of TPS, TDA*, and DMN-ZS*. However, as we presented in our **Response (1)**, our DPE exhibits a performance gain of over 0.8% across 5 datasets compared to these methods.
>
> ---
>
> If you have additional comments or concerns, please let us know.
>
> Best,
>
> Authors

---

> > ### Author Response · Authors · 2024-08-13
> >
> > Dear Reviewer Sqqz,
> >
> > We greatly appreciate the time you have dedicated and the valuable feedback you have provided. As the discussion period draws to a close (Tue, August 13), please kindly let us know if there are any remaining questions. We will be more than happy to provide any further details or clarifications.
> >
> > Best,
> >
> > Authors

---

> > ### Comment · Reviewer_Sqqz · 2024-08-14
> > **Official Comment by Reviewer Sqqz**
> >
> > Thanks for your detailed response. The novelty of the core idea is still my primary concern, and I have a few questions about the response.
> >
> > (a) Which model (DMN-ZS or DMN-ZS*) in response (4) corresponds to DMN-ZS in response (1)?
> >
> > (b) Can you supplement performance comparison based on CLIP-ViT-B/16? You can just combine existing results into one table.

---

> ### Author Response · Authors · 2024-08-14
> **Many thanks for your follow-up feedback**
>
> Dear Reviewer Sqqz,
>
> Thank you for providing further detailed comments.
>
> ---
>
> **Comment (a)**: “*Which model (DMN-ZS or DMN-ZS\*) in response (4) corresponds to DMN-ZS in response (1)?*”
>
> **Response (a)**: DMN-ZS* refers to the DMN-ZS mentioned in **Response (1)**. We evaluated its performance and efficiency using the official code implementation of DMN-ZS, where the pseudo-label predictions are by default enhanced by $N=64$ augmented views and confidence selection. For the DMN-ZS variant in **Response (4)**, we manually set $N=1$ and reported the results correspondingly.
>
> ---
>
> **Comment (b)**: “*Can you supplement performance comparison based on CLIP-ViT-B/16? You can just combine existing results into one table.*”
>
> **Response (b)**: We apologize for not including the full results in the rebuttal post due to character limit. Please find the complete comparisons below:
>
> | Method         | ImageNet  | ImageNet-A | ImageNet-V2 | ImageNet-R | ImageNet-S |  Average  | OOD Average |
> | -------------- | :-------: | :--------: | :---------: | :--------: | :--------: | :-------: | :---------: |
> | CLIP-RN50      |   58.16   |   21.83    |    51.41    |   56.15    |   33.37    |   44.18   |    40.69    |
> | TPT            |   60.74   |   26.67    |    54.70    |   59.11    |   35.09    |   47.26   |    43.89    |
> | DiffTPT        |   60.80   | **31.06**  |    55.80    |   58.80    |   37.10    |   48.71   |    45.69    |
> | TDA            |   61.35   |   30.29    |    55.54    |   62.58    |   38.12    |   49.58   |    46.63    |
> | TPS            |   61.47   |   30.48    |    54.96    |   62.87    |   37.14    |   49.38   |    46.36    |
> | DMN-ZS         | **63.87** |   28.57    |    56.12    |   61.44    |   39.84    |   49.97   |    46.49    |
> | **DPE (Ours)** |   63.41   |   30.15    |  **56.72**  | **63.72**  | **40.03**  | **50.81** |  **47.66**  |
>
> | Method         | ImageNet  | ImageNet-A | ImageNet-V2 | ImageNet-R | ImageNet-S |  Average  | OOD Average |
> | -------------- | :-------: | :--------: | :---------: | :--------: | :--------: | :-------: | :---------: |
> | CLIP-ViT/B-16  |   66.73   |   47.87    |    60.86    |   73.98    |   46.09    |   59.11   |    57.20    |
> | TPT            |   68.98   |   54.77    |    63.45    |   77.06    |   47.94    |   62.44   |    60.81    |
> | DiffTPT        |   70.30   |   55.68    |    65.10    |   75.00    |   46.80    |   62.28   |    60.52    |
> | TDA            |   69.51   | **60.11**  |    64.67    |   80.24    |   50.54    |   65.01   |    63.89    |
> | TPS            |   70.19   |   60.08    |    64.73    |   80.27    |   49.95    |   65.04   |    63.76    |
> | DMN-ZS         | **72.25** |   58.28    |    65.17    |   78.55    | **53.20**  |   65.49   |    63.80    |
> | **DPE (Ours)** |   71.91   |   59.63    |  **65.44**  | **80.40**  |   52.26    | **65.93** |  **64.43**  |
>
> As shown, our proposed DPE still outperforms TPS and DMN-ZS by 0.89% and 0.44% on average across 5 datasets using the ViT-B/16 backbone of CLIP. We have updated the results in Table 1 of the revised manuscript. For more performance comparisons, please also kindly consider referring to Table B in the attached one-page PDF in the general response, where we further compare our method with TDA using the OpenCLIP ViT-L/14 backbone, and observe consistent performance improvements.
>
> ---
>
> Finally, we would like to summarize the key novelties of this work:
>
> - To the best of our knowledge, our work is the first to capture domain-specific knowledge from a **multi-modal** perspective for test-time adaptation of VLMs. Specifically, we achieve this by evolving two sets of prototypes from both textual and visual modalities to progressively capture more accurate multi-modal representations for target classes during test time.
>
> - We proposed textual and visual prototype evolution to extract historical knowledge from previous test samples, enabling **effective accumulation** of knowledge over time.
>
> - We further introduced prototype residual learning with an **alignment constraint** to enhance **consistent** multi-modal representations and ensure alignment of prototypes across modalities.
>
> - Our proposed DPE achieves state-of-the-art performance across 15 various datasets while also exhibiting competitive test-time efficiency.
>
> ---
>
> We hope that our responses have addressed your concerns. Please kindly let us know if you have any further questions.
>
> Best,
>
> Authors

---

> > ### Comment · Reviewer_Sqqz · 2024-08-14
> > **Official Comment by Reviewer Sqqz**
> >
> > Thanks for your response.
> >
> > Most of my concerns about the experiment have been addressed, but the performance improvement (+0.9%/+0.4%) is not very promising considering the increase in inference time (+100%, nearly double).
> >
> > Meanwhile, I maintain my opinion on novelty, so I choose to maintain my score as Borderline accept.

---

### Author Rebuttal · Authors · 2024-08-06

Dear AC and Reviewers,

We are sincerely grateful to you all for dedicating time and efforts in providing these detailed and thoughtful reviews, which helped us to improve the quality of our paper. We also want to thank all the reviewers for your **unanimous recognition and positive recommendations** of this work.

Here, apart from the point-by-point responses to each reviewer, we would like to summarize the contributions of this work and highlight our new results added during the rebuttal phase.

---

We are delighted that the reviewers appreciate and recognize the following strengths and contributions of this work:
- The proposed method addresses a crucial challenge in real-world applications by enabling efficient and effective test-time adaptation without the need for annotated samples from the target domain. **[SVvN]**
- The motivation and idea are very clear and easy to follow, and the method design is reasonable. **[Sqqz, vqJd]**
- The introduction of dual prototypes (textual and visual) for evolving task-specific knowledge at test time is a novel concept. **[SVvN]**
- Comprehensive experiments on 15 various datasets verify that our proposed DPE significantly improves the generalization capabilities of VLMs. DPE also shows improved computational efficiency compared with TPT and DiffTPT. **[Sqqz, SVvN, vqJd]**
- The paper is well-written, the structure is clear and the methodology is well explained. **[All Reviewers]**

---

In this rebuttal, we have included the following discussions and experiments to address reviewers’ comments:
- We discuss the unique contributions of our work compared to DMN-ZS, TaskRes. **[Sqqz, owvr, vqJd]**
- **[Table A]** We provide further performance comparisons with DMN-ZS and TPS to verify the effectiveness of our method. **[Sqqz, vqJd]**
- **[Table B]** We test our DPE method on the larger-scale OpenCLIP model with the ViT-L/14 backbone to verify that our method generalizes well to other VLMs. **[owvr, SVvN]**
- **[Table C-F]** We conduct more detailed ablation studies analyzing different algorithm components, two loss terms, update steps, and hyperparameter $M$, explaining their different impacts in greater detail. **[Sqqz, owvr, vqJd]**
- **[Table G-H]** We offer a more detailed analysis of our computational overhead in terms of both inference time and GPU memory. **[Sqqz, SVvN, vqJd]**

---

Again, thank you for your time in reviewing our work!

Best,

Authors

---

### Decision · Program_Chairs · 2024-09-25

**Decision:**

Accept (poster)

**Comment:**

The paper introduces a novel method for test-time generalization in vision-language models (VLMs), specifically applied to CLIP. The method, DPE, evolves both textual and visual prototypes during test time to capture accurate multi-modal representations. The method aims to improve performance on out-of-distribution tasks by progressively accumulating knowledge from unlabeled test samples. The approach is evaluated on 15 benchmark datasets, demonstrating superior results compared to existing methods.

While the performance improvement is marginal and the complexity increased, the method introduces a novel and technically solid approach to improving VLMs in test-time adaptation settings. The comprehensive experimental evaluation and potential to inspire future research in test-time generalization make the paper a worthwhile contribution to the field, justifying acceptance to NeurIPS.